# Optimal Microbiome Networks: Macroecology and Criticality

**DOI:** 10.3390/e21050506

**Published:** 2019-05-17

**Authors:** Jie Li, Matteo Convertino

**Affiliations:** 1Nexus Group, Graduate School of Information Science and Technology, Hokkaido University, Sapporo 060-0814, Japan; 2GI-CORE Global Station for Big Data and Cybersecurity, Hokkaido University, Sapporo 060-0814, Japan

**Keywords:** microbiome, complex networks, species diversity, criticality, RSA, information flow, transitions

## Abstract

The human microbiome is an extremely complex ecosystem considering the number of bacterial species, their interactions, and its variability over space and time. Here, we untangle the complexity of the human microbiome for the Irritable Bowel Syndrome (IBS) that is the most prevalent functional gastrointestinal disorder in human populations. Based on a novel information theoretic network inference model, we detected potential species interaction networks that are functionally and structurally different for healthy and unhealthy individuals. Healthy networks are characterized by a neutral symmetrical pattern of species interactions and scale-free topology versus random unhealthy networks. We detected an inverse scaling relationship between species total outgoing information flow, meaningful of node interactivity, and relative species abundance (RSA). The top ten interacting species are also the least relatively abundant for the healthy microbiome and the most detrimental. These findings support the idea about the diminishing role of network hubs and how these should be defined considering the total outgoing information flow rather than the node degree. Macroecologically, the healthy microbiome is characterized by the highest Pareto total species diversity growth rate, the lowest species turnover, and the smallest variability of RSA for all species. This result challenges current views that posit a universal association between healthy states and the highest absolute species diversity in ecosystems. Additionally, we show how the transitory microbiome is unstable and microbiome criticality is not necessarily at the phase transition between healthy and unhealthy states. We stress the importance of considering portfolios of interacting pairs versus single node dynamics when characterizing the microbiome and of ranking these pairs in terms of their interactions (i.e., species collective behavior) that shape transition from healthy to unhealthy states. The macroecological characterization of the microbiome is useful for public health and disease diagnosis and etiognosis, while species-specific analyses can detect beneficial species leading to personalized design of pre- and probiotic treatments and microbiome engineering.

## 1. Introduction

### 1.1. Microbiome Dynamics and Health

Microbial ecology has become an important topic for health sciences and other basic and applied sciences such as biology, ecology, forensics and agriculture. In particular, the microbiome seems particularly important for ecosystem health in a broader sense, being the primary connector among multiple species, ecosystem structure, functions and services [1,2]. Recent work has shown how each person maintains a fairly unique microbial fingerprint, and that microbial dysbioses are often associated with shifts in health-status. These shifts are typically associated with the gut that is the most diverse part of the human body considering the bacteria holobiont [3,4]. We recognize that our microbiota is highly dynamic, and that this dynamics is linked to environmental and individual states [4]. The field of microbiome science is still in its infancy and it is not yet settled upon whether gut microbial community structure varies continuously or if it jumps between “discrete” community states, and whether these states are in common across individuals. In particular, some researchers suggest that gut communities can be binned into discrete enterotypes [5], while others argue that gut communities vary along multidimensional continua without any universality [6]. If the ultimate goal of microbiome research is to improve human health by engineering the ecology of the gut, and other applications are also of interest, we must first understand how and why our microbiota varies in time and space, whether these dynamics are consistent across humans, whether we can define stable or healthy dynamics, and how these states are associated to the environment. This line of research is primarily missing how microbial diversity is organized considering all its facets and how this diversity changes when species interaction networks change. For instance, the same level of diversity can be achieved via different network topologies that may lead to different health states [7].

### 1.2. Microbiome Diversity and Functional Network Organization

To determine the network organization of the microbiome and associate that to healthy or unhealthy states, we consider Irritable Bowel Syndrome (IBS) as the template syndrome to characterize microbiome dynamics [8,9]. IBS shows common symptoms of cramping, abdominal pain and diarrhea related to altered gut flora. Previous research has found that the microbiome in people with IBS differs from that in healthy people [8]; however, nobody has demonstrated how the microbiome network is different for these healthy and unhealthy individual groups (i.e., “states” generally speaking when not focused on a particular subpopulation) and how the transition from one to another occurs. By exploring this topic, we propose novel network inferential models for gathering microbiome networks from species big data; these models are based on the principle of maximum entropy that tries to gather the most informative set of variables about stable state patterns with the least amount (but most diverse set) of information [10,11,12,13]. An example can be about sets of species abundance for predicting a diverse set of potential species interaction networks. “Big data” is not only related to the size of the data used but also to the number of calculations required to infer the underlying networks. These computations increase exponentially with the number of species/nodes *n* considered beyond the geometrical criteria, where the number of connections is n(n−1) in the case of an undirected topology of the network. A directed topology is for instance found when species interaction networks are non-symmetrical which means that the direct influence of two species does not have the same magnitude for different directions of interaction [14]. A variety of different models have been proposed to infer network structures from small and large datasets. For biological systems in particular, the inference of causal interactions among systems’ components is a daunting task because not all interactions are known, nor the “true” magnitude of interactions, considering the data used to assess these interactions and the models [15,16]. For instance, microbiome networks are in principle different if the used input data are species occurrence, relative species abundance (RSA), geographic range or other features. In addition, for this motivation, we employed assumption free inference models that consider the whole probability distribution of species dynamics and these models were validated considering their ability to predict population biodiversity patterns over time. We extracted optimal microbiome networks as optimal information networks (OINs) [13] for healthy, transitory and unhealthy groups to investigate general patterns and drivers underlying microbiome stability and the interactions among different species in terms of network topology, magnitude and preferential direction. Additionally, we characterized macroecological functions α-, β- and γ-diversity, which describe the temporal organization of microbiome biodiversity considering time point, intertemporal and total diversity. We show how these functions are related to microbiome network features and different topologies emerge for different diversity/health states. The linkage between microbiome networks and macroecology (in particular information theoretic and biodiversity functions) is unique and offers additional insights into the ecology and the evolution of the microbiome with relevance to ecosystem health.

### 1.3. Microbiome Inference, Neutrality and Criticality

Speculations about the underlying processes of ecosystems’ organization have been made in the past considering diversity patterns and models able to predict these patterns such as neutral models [17,18,19,20], niche models [21,22,23,24,25], and other models such as Lotka–Volterra models based on non-linear ordinary differential equations [26]. Neutral models posit that biological diversity is driven solely by ecological drift without a strong interference of environmental biases that lead to preferential dynamics (“niche”) for some species versus others. Neutral patterns exhibit species indicators (e.g., RSA) of all sizes simultaneously without a preferential size. From neutral to niche states, a critical transition is typically observed where species network organization exhibits scale-free behavior [22,27,28,29,30,31]. This scale-free behavior was thought to occur only at the critical transition point but recent evidence shows that criticality (defined by the scale invariance of ecosystem function reflected by a Pareto distribution) [32] also exists for stable states where system’s component organization is optimal due to optimal information sharing among components and the environment [20,33]. Transitions in network functions are also observed for neural systems where subcritical and supercritical regimes are defined as the ones corresponding to weakly connected random networks and hyperconnected scale-free networks [34,35] that can associate to pathologies. These transitions were previously found for geophysical networks and coupled ecological networks [36,37] for instance, where energy dissipation tends to a global minimum.

Some indications that microorganism cooccurrence patterns are shaped by species interactions that are altered from niche to neutral is available (e.g., [22,38]). This also has conceptual and numerical confirmation when thinking and simulating species that are just responding to local resources and species that are somehow “equal” and responding to fundamental speciation-dispersal processes. The former are interacting more randomly with limited dispersal ranges while the latter are interacting with much larger dispersal ranges. The corresponding probability distributions of species diversity for the former and latter cases are exponential and power-law, respectively, corresponding to random and scale-free species networks. Without introducing any model (but with the knowledge of the underlying potential macroprocesses) these changes in network topologies have been observed for large scale ecosystems [39] and other single population systems where topologies correspond to system’s pathologies [40].

However, these models of microbiome characterization are typically driven by some “hard” assumptions about the species interaction network, which may lead to erroneous conclusions about the predicted patterns: in other words, predictability (under some assumptions) of biological patterns does not imply causality considering the hypothesized and implemented processes [41]. Leaving aside the causality investigation, models of microbiome network inference exist (see, e.g., Baldassano and Bassett [42] and Stein et al. [26]) but they simply infer species co-occurrence networks without assessing the magnitude and directionality of potential species interdependence. A different approach is achieved by pattern-oriented models charactering systems’ dynamics [43,44,45] such as the one here proposed, which do not assume any preferential mechanism a priori but consider the whole information content in data (via probability distributions and their relevance to predict patterns via entropic functions [11]) to claim underlying processes. In this sensem we move our discussion of the problem of understanding microbiome dynamics toward one that identifies which information is critical, and how that model criticality [11,46] is associated to biological criticality [32] also considering the neutrality of biodiversity dynamics. Therefore, rather than trying to untangle biological complexity via fitting some biologically inspired models, we use all data available to check their information content to define all possible microbiome states and associated diversity patterns. In this information theoretic framework, in particular we show how criticality coincides with neutrality and optimal microbial network organization that leads to healthy states. We also show how criticality corresponds to a scale-free functional networks relating RSA interdependencies even when the functional co-occurrence network of species is not scale-free (this place some warning about inferring networks just based on occurrence data).

As a caveat, it should be noted that neutral patterns does not necessarily imply neutral processes [47] despite many papers try to define one from the other [22,23,48,49,50]. Furthermore neutral models can predict non-neutral processes (therefore care must be placed when considering predictability vs. causality) and neutrality might not be present at all scales of biological organization [23]. The focus here is on microbiome pattern detection and its predictability, which we believe to be extremely important and the starting point for a top-down investigation of the underlying processes and causality. Different patterns are evident for different health states when RSA interdependence networks are considered, and these networks seem to shape microbiome diversity in many ways considering local, intertemporal and total diversity.

## 2. Material and Methods

### 2.1. Microbiome Data

We considered microbiome data originally published by Durbán et al. [51] and later used by Martí et al. [8] for which species data of six individuals are available over time (30 days). Fine scale species Operational Taxonomic Unit (OTU) RSA data were derived by published 16S rRNA and shotgun metagenomic sequencing (SMS) data pertaining to the gut microbiotas. In Durbán et al. [51], species-level phylotypes were defined at 97% of sequence identity, which is the lowest taxonomic rank used to identify differences in biological states of interest (e.g., healthy and unhealthy). Two individuals suffered from IBS, two were healthy, one was treated with antibiotics and one was on the verge of being unhealthy. Thus, these two individuals are representative of a transitory state with different directions, from unhealthy to healthy and from healthy to unhealthy, respectively. Durbán et al. [51] considered the healthy subjects as those individuals who did not suffer from lab-confirmed IBS, and took the patients who had this disease as individuals with perturbations from the healthy state without a priori categorization. In the dataset [51], the healthy period is from time points before the IBS triggering event altering the microbiome. More specifically, the datasets are composed by two healthy individuals (Individuals A and B in the original datasets [8,51]), two transitory individuals (C and C1), and two patients with IBS (P1 and P2). The length of RSA data for these individuals are 30 days for A, 15 days for B, 15 days for C, 9 days for C1, 9 days P1, and 14 days for P2.

### 2.2. Time Series Reconstruction

The raw data available present the challenge of individuals whose species abundance is sampled for different time lengths. Computationally, to have datasets with the same length and merge them into one group, we used the method of Least Common Multiple (LCM) [52] for time series reconstruction. LCM extends time series at their maximum feasible length by preserving their probability distribution functions (pdfs); in our case, the pdfs are associated to each RSA and are the inputs for the network inference model that requires time series with the same length [53]. We calculated LCM considering the number of data for each individual in each group. The extended length is the smallest number that is a multiple of the length of original time series of each individual. This implies to extend the time series at the length of LCM or to maintain the data length if the length of the raw data is equal to LCM. In this way, LCM guarantees to have the largest dataset representative of the stochastic dynamics analyzed. In our study, LCM between Individuals A and B was 30; thus, the length of the abundance time series for A was unchanged while B became 30 (B was repeated twice). This was done by copying the data in B until the 30th day. LCM for C and C1 was 45; thus, both C and C1 time series were extended to 45. LCM for P1 and P2 was 126; thus, both time series were expanded to the 126th day. These examples show that data rich sample are preserved as they are while data poor samples are extended. To create pdfs of RSA representative of each group, we considered the average values of RSA for common species. If for individuals belonging to the same group different species were found, the pdf of RSA was based on the time series as they were. This choice was dictated by the desire to emphasize common dynamics for each group when possible.

### 2.3. Probabilistic Characterization of the Microbiome

We characterized probabilistically the distribution of microbiome macroecological and species interaction network variables (generally indicated as *Y* as for a generic random variable) considering the following general exceedance probability distribution function (see Convertino et al. [54]):(1)$$\begin{array}{c} P\left(Y\geq y\right)∼\left\{\begin{array}{cc} e^{-\lambda_{1}\;y} & \mathrm{for}y<Y^{*}\\ y^{-\epsilon +1}\;f\left(\frac{y}{m}\right)\;e^{-\lambda_{2}\;y} & \mathrm{for}y\geq Y^{*} \end{array}\right., \end{array}$$
where $Y^{*}$ is the truncation point (“hard truncation”) for which the transition in the regime of the probability distribution is observed from exponential to power-law. We refer to “hard truncation” when the pdf clearly exhibits two regimes (for $y<Y^{*}$ and $y>Y^{*}$) in which two diverse pdfs can be identified. λ factors are scale factors for the exponential distribution (related to random networks), either above or below the lower/upper cutoff defining the scale-free regime with power-law distribution (associated to scale free networks). *m* is the upper cutoff after which finite size effects occur faster than exponential decays. We introduce the function f(y/m) to give more generality to the cutoff (or homogeneity) function [54]. $y^{-\epsilon +1}$ is the scaling function where ϵ is the scaling exponent of the power-law distribution; this exponent is a critical exponent associated to the fractal dimension of the process analyzed, yet it is representative of the process dynamics [54]. Note that the probability distribution function $p\left(y\right)\;y^{-\epsilon }$ scales with ϵ only. ϵ dictates how the mean and the variance behave, in fact it is related to the Taylor’s law scaling exponent [8]. For ϵ=2, the pdf is the classical Zipf’s law that is found for many socio-ecological systems [54,55].

### 2.4. Network Inference and Dynamical Species Characterization

#### 2.4.1. Information Balance and Exchange

To infer species interaction networks based on microbial RSA data, we based our approach on the model developed in Servadio and Convertino [13] as well as on previous computational efforts [53,56]. We considered the microbiome as a dynamic network of species interactions (sensu RSA interdependence vs. true causality) where the total free energy and corresponding entropy change over time. Codes of the model are available at the GitHub account https://github.com/HokudaiNexusLab/Microbiome. The pdf of each RSA for each group was derived by putting together the RSA time series for all individuals; in this network, the RSA was treated as a random variable meaningful of the group and each individual was offering one realization of the same random variable. The RSA matrix was created with compositions in mind and therefore the sum of each sample was constrained [57]. Considering information entropy as the total dissipated energy’s counterpart, the total network entropy can be written as:(2)$$H\left(N\right)\approx \sum_{i}H\left(x_{i}\right)+\sum_{i}\sum_{j\neq i}TE_{i}\left(x_{i},x_{j}\right)+\sigma \left(N\right)$$
where $x_{i}$ denotes the i−s variables that contribute to the total information of the network *N*. In our case, *x* is the RSA of species. In this equation, $H(x_{i})$ denotes Shannon entropy, and $TE(x_{i},x_{j})$ denotes Transfer Entropy from the first variable to the second variable [13,56,58,59,60]; in our case, both variables are the RSA of two different species. Equation (Equation 2) represents a fundamental principle of information balance independently of the chosen entropy analytics [61] and forms the general basis of sensitivity analyses. Equation (Equation 2) states that the total network entropy can be decomposed into the entropy of each individual node plus the entropy of interactions. The sum of absolute TEs is a proxy of the Mutual Information (MI) of a variable, thus it considers the whole set of variable interdependencies; in Equation (Equation 2), we consider the sign of TE because H(N) should consider the typology of interactions with their sign. σ(N) is a noise term that captures the unexplained variability of *N* related to variables not considered and other discretization factors related to the numerical methods employed in solving the model. Shannon entropy is representative of the species information content (attached to the pdf of RSA) for the whole network and it allows comparing all species in a common framework. Equation (Equation 2) can also be extended in space if spatially explicit calculations are needed, as in Servadio and Convertino [13]. Note that H(N) is inversely proportional to the free energy of the system so the lower H(N) the higher the free energy and the higher the total dissipated energy. Evolution self-organizes systems toward states where H(N) is minimized [10,33].

The computation of TE was based on the distributions of the two variables of interest (i.e., RSA) conditioned on their histories. Comparing the conditional probability of the variable on its own history with the conditional probability of the variable on both its own history and the history of a predictor variable provides asymmetry in determining predictive abilities of one variable onto another. Thus, a directed network can be inferred. Directed TE of two time series variables, denoted as $X_{i}$ and $X_{j}$, was calculated as
(3)$$TE_{X_{i}\;\to X_{j}}=\sum p\left(X_{j,t},X_{j,\tau },X_{i,\tau }\right)\cdot \log\left(\frac{p(X_{j,t}|X_{j,\tau },X_{i,\tau })}{p(X_{j,t}|X_{j,\tau })}\right)$$
where $X_{i,\tau }$ and $X_{j,\tau }$ denote the respective histories of $X_{i}$ and $X_{j}$ at time *t* as well as considering all past values for the period t−τ. Here, we consider the same memory lag for $X_{i}$ and $X_{j}$ but in principle historical dependencies can be different when considering other variables and the variable itself. In our microbiome study, $X_{i}$ and $X_{j}$ are RSA of species *i* and *j*. This definition is the most general definition of TE and neither conflates dyadic and polyadic relationships between species nor assumes any causality [62].

The definition of TE can assume that the processes analyzed obeys a Markov model, which is suitable for memoryless stochastic process. This implies that future states depend only on the current state and not on events that occurred before it. Thus, in a Markov process, it is assumed that τ=1. This is usually true, especially for rapidly varying processes (such as for microbial RSA); however, this constraint can be relaxed by choosing temporal lags that are small enough to focus on short-term interdependencies which are not related to long dependencies in the underlying processes. In our case, study RSA values of two randomly selected species did not correlate with RSA values for τ=1; thus, memory processes are relevant and, as in Villaverde et al. [53], we selected the τ that maximizes the interdependency between two species assessed by the functional distance (see Equation (Equation 12)). Note that TE, as calculated in Equation (Equation 3), should be interpreted as information flow vs. information transfer (as in Lizier and Prokopenko [15]) because conditional entropies are used to exclude indirect pairs of species whose interactions is of second order importance. This approach has been criticized by some authors (e.g., James et al. [62]) if “causality” is indeed claimed about the inferred interactions and in consideration of the fact that polyadic relationships may be underrepresented. In this study we spouse the view of James et al. [62] for which TEs are considered as measures of reduction in uncertainty about one time series given another (thus, with predictable power) with potential but not certain causality, leaving aside the issue of what specific biological causality is investigated (e.g., influence, physical causality, etc.). The idea of using conditional entropies is solely related to find the most informative set of species to identify the core microbiome interaction network.

#### 2.4.2. Maximum Entropy Networks

Subsequently, the inference of interspecies TEs, among all values of TEs the question remains on which value is the most informative about the potential causal relationship between two variables. We emphasize that here “causal” is in the sense of of predictability, sensu uncertainty reduction, rather than “certain” biological reality. As in Servadio and Convertino [13], we proposed to select TEs that lead to the maximum entropy for the inferred network. This corresponds to maximize the Fisher information matrix [63] that produces the lowest complexity and the highest informative set of information about a pattern of interest. MaxEnt [12] favors probability distribution functions with maximum entropy as the most general distributions that fit the observed data [64]. This theory can be applied to a functional network where edge weights are based on TE. The network with the greatest total entropy can be similarly favored as the most general network structure that fits the observed data. The method considers all possible pairs of variables in both directions for predicting a pattern of interest. The edges that comprise the network with the greatest total TE are then included. Selecting the edges that contribute to the greatest amounts of TE, according to the MaxEnt theory, produces the network that most accurately describes “causal” patterns among the included variables. Note that MaxEnt should be interpreted in an information theoretic sense, where higher entropy means higher information. We show how this entropy (useful to characterize the system) is related to the state of each health group that has a more ecological and physical sense in a thermodynamic purview; in particular, how the absolute value of total entropy is lower for stable and healthy states vs. unhealthy ones.

A utility function is needed to establish the function where MaxEnt is applied. The utility function can be thought as a systemic (network) value function $\sum_{i,j}\;f_{i,j}\left(X\right)\;w_{i,j}$ (potentially multiplied by weight factors $w_{i,j}$) where value functions $f_{i,j}$ are TEs among RSAs. These TEs, as in Equation (Equation 3), assess the potential causal interactions between species pairs. Thus, the utility function is the total network entropy H(N) (Equation (Equation 2)) that needs to be optimized in order to define necessary and sufficient TEs with the maximum entropy. The optimization can be subjected to feasibility constraints, for instance related to the ability to control certain species or data limitations. In the context of the present goal of creating a microbiome network indicator, the value functions $f_{i,j}$ are defined as:(4)$$\begin{array}{c} f_{i,j}\left(X\right)=\left\{\begin{array}{cc} TE_{X_{i}\;\to X_{j}}, & \mathrm{for}\left\{X_{i},X_{j}\right\}\in E_{MENet}\\ 0, & \mathrm{for}\left\{X_{i},X_{j}\right\}\notin E_{MENet} \end{array}\right., \end{array}$$
where $\left\{X_{i},X_{j}\right\}$ represents the directed edge connecting $X_{i}$ to $X_{j}$, and MENet (Maximum Entropy Network) represents the set of directed edges in the network with the maximum total network entropy H(N). The selection of edges to be included in the network is determined by finding the network with the greatest total entropy as in Equation (Equation 2). In the present study, the utility function was defined as the total TE of the network (plus Shannon entropies of each RSA but those turned out to be second- or third-order factors that can be neglected), and it is maximized by selection of the $f_{i,j}$ functions. To the best of our knowledge, this is one of the the first times that TE was framed in a decision analytical model via a network threshold entropy criteria that defines MENets.

#### 2.4.3. Optimal Information Networks

To reduce redundancy in creating a MENet, variables that are strongly predicted by other variables (hypothetically establishing a strong causality—in a predictive sense rather than in a biological one—if prediction accuracy of one decreases quickly when removing the other [41]) can be excluded. This can be done by evaluating the weighted in-degree and out-degree of each node in the network (i.e., TE). Nodes with a greater weighted out-degree than in-degree can be included in the Optimal Information Network (OIN) that one among many MENets with the same average total entropy. These nodes are strongly predicting the variability of other nodes, thus the overall network dynamics. OIN is then the necessary and sufficient MENet for predicting microbiome function. Here, we refer to microbiome function as the information network related to the interdependence between RSA measured by TE; this function is not the “true” biological function but it is likely related to the variability in mutual abundance that is commonly found in any complex ecological systems [65,66]. Thus, OINs are purely information networks and not causal biological networks. This entropy reduction to define OINs based on conditional entropies (calculated on sets of potentially influencing species that do not affect much the total entropy, yet removing the indirect interactions as in Lizier [56] in order to estimate information flow vs. information transfer [15], where the former is more likely representing “causal” species interactions)) can be further achieved by introducing functions $g(X_{i})$, defined as follows
(5)$$\begin{array}{c} g\left(X_{i}\right)=\left\{\begin{array}{cc} 1, & \mathrm{for}\sum_{j}f_{i,j}\left(X\right)>\sum_{j}f_{j,i}\left(X\right)\\ 0, & \mathrm{for}\sum_{j}f_{i,j}\left(X\right)\leq \sum_{j}f_{j,i}\left(X\right) \end{array}\right., \end{array}$$
where $\sum_{j}f_{i,j}\left(X\right)=OTE$ and $\sum_{j}f_{j,i}\left(X\right)=ITE$. OTE and ITE are the total outgoing and incoming TE for a node, respectively. Thus, variable inclusion depends on the comparison of the TE projected by the variable $X_{i}$ onto the other variables and the TE projected by the other variables onto $X_{i}$.

The defined function *g* was then used to create the total network entropy that can be used to carefully describe the network dynamics:(6)$$H\left(N\equiv OIN\right)=\sum_{i}H\left(x_{i}\right)\cdot g\left(x_{i,t}\right)+\sum_{i}\sum_{j\neq i}TE_{i}\left(x_{i},x_{j}\right)\cdot g\left(x_{i,t}\right)+\sigma \left(Y\right)$$
which represents the sum of all necessary variables that were included by the structure of MENet in a multi-criteria value function, and the sufficient variables after the redundancy exclusion to form OIN. In this way, the OIN inference was based on information theoretic and functional topological criteria to screen: (i) the necessary information to maximize network entropy H(MENet) (i.e., total information content); and (ii) the smallest non-redundant information to sufficiently predict total network function (of maximum entropy H(OIN)). Note that the first criterion on H(MENet) is a global one on the total information content while the criterion on H(OIN)) is a local one on the information of a node with respect to the functionally connected nodes. This entropy minimization is somehow the equivalent of the energy minimization of other optimized networks in nature [67].

However, this OIN is the network with the highest accuracy in predicting macroecological patterns of diversity over time that are dependent on fluctuating RSA. Then, OINs are characterized by the highest information content (lowest uncertainty), highest information diversity (e.g., represented by the values of TEs), and lowest complexity.

#### 2.4.4. Assessment of Species Importance and Collectivity

After the inference of OINs, it is possible to quantify the importance of different species considering their variability in isolation and in cooperation with other species for predicting the dynamics of the microbiome. Species first order importance and interaction for reproducing the network dynamics are then calculated considering new indices based on nodal information flow rather than on Mutual Information Indices (MII) as in Lüdtke et al. [68]. $\sigma_{i}$ describes species interaction and is calculated as the ratio between the total Outgoing Transfer Entropy (OTE) as information flow ($OTE\left(j\right)=\sum_{i}TE_{j\to i}$) and the total network entropy, while $\mu_{i}$ describes the species importance as the ratio between the nodal Entropy as information content (using Shannon entropy) and the total network entropy. These Transfer Entropy Indices (TEI) are useful when no systemic variable is needed (contrary to Servadio and Convertino [13]), and analytically they are formulated as:(7)$$\begin{array}{c} TEI=\{\begin{array}{c} \sigma_{i}=\frac{OTE\left(j\right)=\sum_{i}TE_{j\to i}}{H(OIN)}\\ \mu_{i}=\frac{H\left(x_{i}\right)\cdot g\left(x_{i,t}\right)}{H(OIN)}, \end{array} \end{array}$$

When considering a systemic indicator (see, e.g., Servadio and Convertino [13]), MII are better suited to identify variable importance because no directional influence is needed. MII use the mutual information (MI) normalized by the entropy of the output variable considering one independent variable or pairs of variables for predicting a dependent variable *Y* that is in this case undefined. These MII indices are $s_{i}=\frac{MI(X_{i};Y)}{H(Y)}$ and $s_{ij}=\frac{MI(X_{i};X_{j}|Y)}{H(Y)}$, where $X_{i}$ is any variable (e.g., RSA) and *Y* is the predicted variable built using the same process of constructing OINs but selecting variable features rather than keeping entropy of species as independent variables. The use of TE can give further information about the directionality of causality (in a predictive sense of the model), and the time-lag of the causality.

### 2.5. Macroecological Indicators

To characterize the microbiome as an ecosystem we introduce macroecological indicators that aim to describe ecosystems’ collective dynamics of diversity locally, within communities or time points, and globally. In this paper we use such macroecological indicators that are time dependent (because space information is not provided and hardly inferable) and of order zero mathematically speaking (as in Jost [69] the order is related to the exponent to which the probability of RSA is elevated to). For a set of unique distinct species $S=\{S_{1},S_{2},\ldots ,S_{n}\}$ whose RSA $X=\{X_{1},X_{2},\ldots ,X_{n}\}$ changes over time, we define the local species diversity, or α-diversity as:(8)$$\alpha \left(t\right)=\sum_{k=1,t}^{n}p_{k}{\left(t\right)}^{0}$$
where $p_{k}\left(t\right)$ is the probability to find one species at time *t*. Thus, α is the sum of diverse species at any given time during the observation period (30 days) or the reconstructed period (see Section “Time Series Reconstruction”). Considering this definition of α it is easily noticeable that the sum of the entropy of all RSA $H_{\alpha }=\sum_{k}H\left(x_{k}\right)=-\sum_{k}p_{k}\left(t\right)\;\log\;p_{k}\left(t\right)$ is proportional to the Shannon index that is the local species diversity of order one [69].

Leaving aside the controversy about the definition of interspecies diversity over time, i.e., species turnover, we define β-diversity as the complementary variable of species similarity (here introduced via the Jaccard Similarity Index (JSI) as in Convertino et al. [37] and Convertino [18]):(9)$$\beta \left(t\right)=1-JSI\left(t\right)=1-\frac{S_{t,t+1}}{S_{t}+S_{t+1}-S_{t,t+1}}$$
where $S_{t,t+1}=\sum_{k=1,t}^{n}\left(p_{k}{\left(t\right)}^{0}+p_{k}{\left(t+1\right)}^{0}\right)/2$ is the number of species present at both time steps if $p_{k}{\left(t\right)}^{0}$ and $p_{k}{\left(t+1\right)}^{0}$ are ≠0, otherwise $S_{t,t+1}=1$. $S_{t}=\sum_{k=1,t}^{n}p_{k}{\left(t\right)}^{0}=\alpha \left(t\right)$ is the number of species present at time *t* (or t+1) (Equation (Equation 8)). Note that, β-diversity as a measure of species turnover overemphasizes the role of rare species as the difference in species composition between two communities or two time steps is likely reflecting the presence and absence of some rare species in the assemblages.

Note that the definition of β in Equation (Equation 9) is proportional to the “true” β that is classically defined as the number of diverse species between two samples (either over space or time). β-diversity can also be defined as a second order index where the entropy related to β is $H_{\beta }=H_{\gamma }-H_{\alpha }$ [69] where $H_{\gamma }=H\left(N\right)$ is the total network entropy (Equation (Equation 2)). Considering the variation of diversity over time β-diversity is proportional to the complementary of the mutual information $1-MI_{X_{i},X_{j}}=1-\sum p\left(X_{j},X_{i}\right)\cdot \log_{2}\left(\frac{p(X_{j},X_{i})}{p\left(X_{j}\right)\;p\left(X_{i}\right)}\right)$. However, 1−β(t) is proportional to the sum of the TEs. These relationships between information theoretic quantities and macroecological indicator is novel and worth being addressed in further papers.

The total diversity γ is defined as:(10)$$\gamma \left(t\right)=\sum_{k=1,t=1}^{S,T}p_{k}{\left(t\right)}^{0}$$
that can be established over time or over the total number of speciation events *M*. *M* is the sum of all species at any given time independently of their diversity calculated from time t=1 to the final time of observation *T*; equivalently, *M* is the number of events when new or existing species are introduced. A speciation event is an event when a species is introduced in the microbiome; this species can be already present or can be a new distinct species that is established over the total number of speciation events *M*. The concept of speciation event is introduced because that determines the number of total species introductions independently of the true temporal dimension. Thus, the speciation event focuses on the dynamics of the process independently of time because it counts events. Considering *M* allows one to map how the total diversity changes as a function of biodiversity meaningful scales, equivalently to the species–area relationship [70].

vs. mapping its change over time (that may not be an influencing variable). The total number of speciation events can be related to the number of unique species *S* (i.e., all distinct species occurred in the time period) as follows:(11)$$M=\sum_{k=1}^{S}m_{i}x_{i}^{0}$$
where *S* is the number of unique species across the whole observation period, $x_{i}$ is the RSA of the counted species, and $m_{i}$ is the number of times that species occurs. Considering the validity of the information balance equation (Equation (Equation 2)) that leads to the diversity balance equation $H_{\gamma }=H_{\alpha }+H_{\beta }$, the total diversity can also be calculated as γ=α·β [69].

### 2.6. Functional and Structural Network Metrics

The topological organization of the microbiome is characterized via structural and functional complex network metrics. Functional metrics are based on information theoretic functions that quantify the interactions among species while structural metrics are based on the geometry of the network and can be derived from the former ones.

The functional distance between species is defined as:(12)$$d_{f}\left(X_{i},X_{j}\right)=\min_{\tau }\;\;e^{-MI(X_{i}\left(t\pm \tau \right),X_{j}\left(t\right))}$$
where the minimum value of the distance is taken for all possible time delays τ. $X_{i}$ and $X_{j}$ are the RSA of species *i* and *j* and MI is the mutual information evaluated for different values of the temporal scale of species dependency τ. The τ that minimizes the distance $d_{f}$ is chosen for capturing the maximum interdependence $MI_{max}$. Such distance as in Villaverde et al. [53] quantifies the magnitude of the most meaningful interactions between species in a predictive sense: the higher MI the shorter the distance that signifies high levels of interaction (sensu predictability) without specifying the directionality. Thus, because of the inability of assessing the direction of interdependence between species (whether that is information transfer or flow [15]), MI (or $d_{f}$ equivalently) is a metric useful for identifying the most interacting pairs of the microbiome rather than individual species.

The calculation of the structural distance is based on the functional distance and the concept of the shortest path. The structural distance is then defined as the minimum number of steps from one node (species) to another independently of the magnitude of these steps (e.g., in terms of TE). Thus, analytically the structural distance is defined as:(13)$$d\left(X_{i},X_{j}\right)=\mathrm{argmin}\left[\sum_{i,j}d_{f}{\left(X_{i},X_{j}\right)}^{0}\right]\;\;\;\;\;\mathrm{if}A_{ij}=1$$
where $A_{ij}=TE_{ij}^{0}$ is the adjacency matrix that can be formulated in terms of TE. The rationale for considering the shortest paths is related to the exponentially large ensemble of distances as a function of the number of nodes and the fact that biological systems always optimize information transmission [67]; however, Pareto shortest paths are always chosen [67,71].

In terms of connectivity, the functional degree is defined for the directed network as the sum of the weighted in- and out-degree (i.e., TE) elevated to a power exponent equal to zero. Then, analytically the functional degree is:(14)$$k_{f}=k_{in}+k_{out}=\sum_{i,j}\left[f_{i,j}{\left(X\right)}^{0}+f_{j,i}{\left(X\right)}^{0}\right]$$
where $\sum f_{i,j}\left(X\right)=TE_{ij}$ is the transfer entropy as defined in Equation (Equation 3).

The structural degree is defined by thinking the network as an undirected network (without signs related to TEs), thus
(15)$$k=\sum_{i}a_{i,j}$$
where $a_{i,j}=1=TE_{i,j}^{0}$ if *i* and *j* are connected. Classically, the structural degree considers the number of connections independently of the bidirectional pathways implied by TE. Thus, functional degree is always greater or equal to structural degree.

## 3. Results

The simplest analysis of the microbiome starts by looking at the temporal trajectories of RSA. By a simple cursory analysis, it was evident that the average RSA of the healthy microbiome is lower than the average RSA of the unhealthy microbiome independently of the species; however, the maximum RSA was found for the healthy microbiome and the species with the highest RSA is one of the the most beneficial for health. A recent dataset with absolute abundances suggests that healthy gut microbiota have higher total abundances than diseased ones [72] but no studies exist about the universality of this abundance-health relationship. By looking into species diversity (Figure 1A), it was observed that the average number of species at any time point (α) is lower for the healthy microbiome than the unhealthy one. This may seem in contrast with previous findings that report higher diversity for healthy microbiome or in general for healthy ecosystems [28,73,74]. A controversy on the subject is already found in literature [73], thus just maximizing total diversity without considering how that diversity grows and is organized is not intuitively a necessary and sufficient ingredient to achieve a stable healthy state [75]. More importantly, the RSA-rank pattern (Figure 1B) shows only one dynamical regime, corresponding to the common Zipf–Mandelbrot model for RSA [76], for the healthy microbiome vs. two regimes for the transitory and the unhealthy microbiomes (double Pareto, lognormal or exponential regime). Figure 1C shows that the decay in richness over RSA is higher for the unhealthy microbiome; this result underlines the fact that higher diversity does not imply stability because of the suboptimal, yet unsustainable distribution of species in the unhealthy microbiome. Stability is related to network topology [3], which also affects diversity [77,78] and the systemic fluctuations of the microbiome, as shown by the Taylor’s law [8] that highlights how variance in RSA abundance changes with the mean. “Optimal” organization is in this case referring to the healthy state as a reference state because it has the smallest fluctuations for the highest achievable total diversity growth rate $\gamma^{\prime }$ (this is the Pareto solution) and the associated network topology is more resilient to random node removal (Appendix A). We will show the Pareto solution has the larger diversity growth rare and a Pareto-like network. Figure 1B,C shows the RSA-rank plot and the Preston’s plot [70] of species diversity dependent on RSA. The RSA-rank shows two dynamical regimes for the unhealthy and transitory groups: a result that likely confirms the bimodality in local species richness α. By plotting the Preston’s plot in log-log, a scaling relationship was found showing a faster decay in species richness for the unhealthy group.

Considering the RSA of species in time, from the most to the least relatively abundant, a transition in the epdf of RSA was observed from a pseudo-normal distribution (corresponding to a homogenous spatial distribution) to a Dirac-like distribution (corresponding to a singular point distribution) considering the maximum and minimum RSA. Appendix A shows the epdf of RSA for the top 10 highest RSA, intermediate 10 RSA, and the least 10 RSA species. the transition is less dramatic, from an exponential to a log-normal-like distribution. Intermediate RSA species, independently of species belonging to the healthy, unhealthy or transitory group, show a scale-free like distribution underlying the fact that these species are fundamentally important in the function of the complex microbiome as highlighted in Lahti et al. [28]. Rare species seem also to display a truncated scale-free behavior (limited by their maximum RSA as a finite size factor rather than limited by spatial biological constraints), which also underlines their importance for the microbiome organization. These pdfs are a signature of species interaction networks for different RSA groups: pseudo-random, scale-free, and small-world topology for the highest, intermediate and lowest RSA class, respectively. Further results discuss the connection between RSA and species information flow.

The inferred microbial networks corresponding to the three microbiome groups are shown in Figure 2 (right plots from top to bottom for the healthy, transitory and unhealthy groups). Maximum entropy networks evidence the different topology in microbiome organization for healthy, unhealthy and transitory group. In the structure of these networks, the size of each node is proportional to the Shannon entropy of the species and the color is proportional to the structural degree. In Appendix A, we show the networks whose nodal color is proportional to the total outgoing TE (OTE) that is likely more representative of node activity in a collective network sense. The higher is the value of the structural degree (or OTE in Appendix A), the warmer is the color. The width of each edge is proportional to the TE between pairs and the direction is corresponding to the directional influence. All OINs are special MaxEnt networks, i.e., networks for which the total network entropy is maximized (MENets) and where redundant nodes are removed (see Section 2.4.3). Thus, OINs allow one to identify the fundamental functional species interactions useful for predicting microbiome dynamics. The transition in network topology, from random to small-world (tending toward a scale-free network) for the unhealthy and healthy groups, is manifested also by the shift in total entropy pattern (left plots in Figure 2 from top to bottom). The latter is asymmetrical and symmetrical for the random/unhealthy and scale-free/healthy microbiomes, respectively. This type of network transitions has been observed for large ecosystems (e.g., Winemiller [79]). The network entropy plots show that network entropy over information flow is roughly symmetrical for healthy individuals, expressing that the interconnectedness in healthy communities is more dynamically balanced than unhealthy ones. Appendix A shows microbiome networks for a high value of the threshold on $TE_{ij}$, which establish the information exchange (of flow) between species above which links become relevant. However, these networks are no more OINs. Considering the total network entropy and its decomposition, it was observed that the most important nodes in terms of OTE (Equation (Equation 6) and Appendix A), that is the information flow necessary to predict all other nodes’ dynamics, are the dominant species in making up the total information network (Appendix A). In other words, the entropy of each single node in isolation $H(x_{i})$ is a second- or third-order factor in determining the total network entropy. Appendix A shows that most species interactions (TEs) are positive for the unhealthy microbiome, which is underlying the evidence that mutualistic positive feedbacks leads to instability; therefore, higher α and γ diversity in short and long term do not guarantee stability if interactions are predominantly in one direction. The healthy microbiome instead has balanced positive and negative interactions that lead to microbiome stability.

Figure 3 shows macroecological indicators of diversity of the microbiome for healthy, unhealthy and transitory individuals. We show that species diversity α, and total species diversity γ are the highest in the unhealthy group (for which average RSA is also the highest) but species similarity 1 −β and the the diversity growth rate $\alpha^{\prime }$ over time are the highest for the healthy group. This is a critical result that shapes microbiome organization around healthy or dysbiotic states. The highest fluctuations in RSA and macroecological indicators (in particular, α and γ) were observed for the transitory and unhealthy groups. These results underline the potential conclusion that too high levels of diversity are possibly unsustainable, leading to unhealthy unstable states related to the abnormally excessive multiplication of species in the gut ecosystem. These species may be invasive from outside sources or subspecies created within the gut as a response to external stressors. It is interesting to note that the behavior of the pdf of α informs about the potential states of the microbiome in each group. The pdf is platykurtic multimodal for the unhealthy microbiome, which suggests the presence of multiple unstable states, and it is leptokurtic monomodal for the healthy microbiome which implies one stable state. The transitory microbiome shows an almost symmetrical pdf that underlines the fact it exists in between the healthy and unhealthy microbiome. These results highlight the resilience of the microbiome as a whole dictated by the ability to change as a function of external stressors as well as the higher stability of the optimal healthy state. However, the latter seems easy to perturb considering the lower entropy (and probability, or corresponding high free energy) defined in one state. This ability to change state is also a good indicator of gut adaptability and human body resilience.

Species collective interaction and singular importance are shown in Figure 4 by plotting the information theoretic TEI $\sigma_{i}$ and $\mu_{i}$ (see Methods, Section “Assessment of Species Importance and Collectivity”, i.e., Section 2.4.4). The top 10 interacting species are also the least relatively abundant for the healthy microbiome and the most detrimental; however, these species are controlled by other species and the microbiome is organized into a healthy state. Appendix A shows that from the top to the least 10 TE species there is a shift in the pdf of RSA from a bimodal to a monomodal distribution for the healthy microbiome. For the transitory and unhealthy microbiome, instead, there is a shift from a leptokurtic (Dirac-like) to a platykurtic pdf (uniform-like). The top 10 TE species are the most detrimental bacteria (“antibiotic”) but their RSA is small for the healthy microbiome; this means that these bacteria are controlled (in terms of RSA variability) by all other beneficial bacteria. The top 10 TE species are mostly characterized by positive interactions (positive TEs) while the least ten 10 TE species are characterized by negative interactions (feedbacks). For characterizing species collectivity or single species dynamics, as well as for predictability, OTE that is a node function is better suited than TE that is a link function. The pdfs of OTE in Appendix A show more clearly the changes in species dynamics for each health state and overall species activity manifested by the magnitude of OTE. The top 10 OTE species are always characterized by positive feedbacks vs. the least 10 OTE species with negative feedbacks (top and bottom plots of Appendix A). Appendix A, by plotting the pdf of all TEs and OTEs for any group, further emphasizes the fact that there is a positive bias and an asymmetry for the unhealthy group species interactions.

The non-linear duality between microbiome structure and function is shown in Figure 5 where structure is considered via the network degree (Appendix A) and function is about the nodal information flow OTE. The epdfs show how microbiome function is much more suited to show functional network topology versus microbiome structure. Function is a much more important property than structure which is just based on geometrical analyses of cooccurrence species networks (e.g., as in Baldassano and Bassett [42]). This scale-free function may be related to the scale-free behavior of the intermediate RSA species, as shown in Appendix A. The Pareto solution has the largest diversity growth rate and is not by chance accompanied by a Pareto-like species interaction network where interactions are inferred by TE (Figure 5B). As shown in Figure 2, visually, the healthy microbiome functional network is tending toward a scale-free topological organization. Statistics of the functional scale-free network based on TE are in Figure 5. This mild scale-free organization (see, e.g., [80], where the authors highlighted the difficulty in defining the classification for these networks into one topology radically) does not correspond to a scale-free distribution of α-diversity (Figure 5C) that instead is exponential. Additionally, some functional network features beyond the inferred RSA-based interdependence (TE and OTE) show a bimodal or Poisson distribution (Appendix A) characterizing more small-world networks rather than scale-free ones. However, we point out how these features are more structural than functional (see Equations (Equation 12) and (Equation 14)) since they characterize species interactions directly. The non-linearity among structure, function and microbiome service (i.e., diversity in this paper) is highlighted when plotting α dependent on functional network degree and distance (Appendix A). α diversity increases for high values of the functional degree (Equation (Equation 14)) but does not have a clear trend when considering the functional distance (Equation (Equation 12)). $\alpha (d_{f})$ is lower for the unhealthy than the healthy microbiome for the same range of functional distances which highlights the more random distribution of diversity in any dysbiotic state. We observed 72, 378, and 9647 unique values of functional distance for the healthy, transitory and unhealthy group. The highest diversity in functional distances for the unhealthy group confirm the fact that the unhealthy microbiome is more densely connected and the number of small distances (high species interdependencies) is lower than the healthy one. However, the healthy microbiome is more clusterized into species clusters. The values of functional distance were normalized and the distribution of α over the normalized distance shows a random arrangement for the unhealthy group with respect to the healthy one (Appendix A).

We found the most interesting results when we combined microbiome service and function indicators, for instance considering total macroecological diversity γ and OTE. Figure 6 shows the relationship between γ and the temporal sampling scale (i.e., the number of speciation events) in analogy to the species–area relationship widely used in macroecology [70]. The plot shows a scaling relationship valid for two orders of magnitude whose exponent is higher for the healthy than unhealthy group underlying the optimal growth of diversity for the healthy microbiome. Considering this optimal diversity growth relationship, it is meaningful how the transitory microbiome has the largest value of $\gamma^{\prime }$ leading to a change in diversity from the healthy species “poor” to the unhealthy species “rich” microbiome. These results are in synchrony with the power-law decay of species similarity 1−β over time (Figure 6C). When considering OTE of species as a function of their RSA, we found a surprising scaling law over four orders of magnitude; this law with an average exponent close to 1/4 (very common in biology, for instance the mass-specific Kleiber’s law [81]) implies a decay in species interaction for highly relatively abundant species. When comparing γ over OTE (Figure 6D), a non-linear growth is detected where a common increase in total diversity occurs until a critical species interaction value, above which γ slows down or remains stationary, at least for the healthy and transitory groups. For the unhealthy group, the growth of γ seems to slow down but not reach a stationary state; this may relate to the continuous multiplicative generation of detrimental species in the gut.

## 4. Discussion

We employed an information theoretic model for the inference of microbial species interaction networks based on RSA interdependence. The model was used to infer microbial networks associated to different health states and is suitable for predicting selected biodiversity patterns characterizing the space-time organization of bacteria α-, β-, and γ-diversity. Thus, the primary purpose of the model is not to infer causal (or “true”) species–species interactions among bacteria. The computational inference of “real” interactions is always very hard—provided that there is a complete knowledge of the reality on which results can be validated—and any inferred interaction is always dependent on the analytics and data used. For instance, RSA profile may not necessarily contain the information about all species–species interactions aimed to be assessed but still the question remains about what is truly an interaction (aimed to be measured) since any physical or functional interaction may not necessarily reflect any change in RSA, or other biomarker. Additionally, any change in RSA or other biomarkers may be related to other external factors, such as environmental fluctuations, which alter species simultaneously. What is certainly true, however, is that, if the inference model detects strikingly different patterns for different population groups, then those patterns likely tell something meaningful about different dynamics and collective environmentally driven changes [43,44]. In this perspective the entropy-based model is focused on the predictability of patterns vs. causal investigation of mechanisms. The proposed model can be applied to both abundance and RSA, or other biomarkers, without any special modification. Theoretically, the pdf of abundance and relative abundance is the same leaving aside numerical artifacts; independently of this, RSA seems better suited for this type of ecological analyses because it informs about changes of species abundance with respect to the whole community. Abundance and/or RSA seems also the most likely to detect species functional roles and interactions as highlighted by recent studies [65,66]. Constructing a network for each health group is the purpose of studies such as ours that try to identify common group dynamics in populations independently of individual variability (see, e.g., Bashan et al. [82], where universal group dynamics in microbiome is the core quest). The identified network topologies have a correspondence with the dynamics of RSA, that is a critical dynamics for the scale-free information network associated to the healthy state, and exponential dynamics for the random network associated to the unhealthy state. The total network entropy is the lowest for the healthy microbiome for any threshold of the information flow TE (Figure 2). This implies higher free energy available to the healthy microbiome and lower information needed to function where information entropy in the physical space can be thought of as the average interspecies communication/interdependence. The lower entropy in species collective interactions has certain implications for data collection, potentially implying fewer data are needed for characterizing healthy microbiomes. This is because one single globally stable state was identified for the healthy microbiome (in the entropy pattern in Figure 2) vs. multiple stable states for the unhealthy microbiome (one globally and two locally stable state for high, medium and low value of network entropy, respectively). These states correspond to different biodiversity states in terms of α, β and γ. The existence of multiple dysbiotic states seems to confirm the previously observed “Anna Karenina effect” [83] where “all healthy microbiome look alike, instead each unhealthy microbiome is diverse in its own way”. More theoretically speaking, the lowest entropy across the system’s landscape of potential states is a sign of criticality that is the state toward which any ecosystem tends to [33]; the critical state is where there is a balance of system’s self-organization and environmental influence [44].

The inferred patterns in this paper are representative of confirmed health states where individuals are confirmed representative samples (Durbán et al. [51] published the original dataset) for IBS and non-IBS people, as reported by Martí et al. [8]. Patterns and methods are proposed to highlight what is relevant to look at when describing state transitions and characterizing health states. The number of individuals sampled in a population matters as a function of expected or reported patterns’ changes. Reliability is not only dependent on the sample size but also on the consistency and differences within and among samples. In this particular study, we found striking differences between potential health states and many times concordant with the reported literature. Further research is required to test the biological universality [82] or local specificity of these patters across a much larger population sample than the one considered. Analyses were made considering varying data lengths for individuals, which did not change any pattern considered significantly. This means that the dynamics represented in the time series is well contained at least in the smallest data sample available. The smallest reliable sample is for ten data points that seems in this case the minimum data length to have in order to have representative probability distributions.

Considering the issue of compositionality, which is related to the issue of having samples consisting of proportions of various species with a sum constrained to a constant [57,84,85], the theory suggests that a small number of species should increase compositional effects. In our case, the number of species is 47 at minimum and that should limit the effect of compositionality because the sample is large enough. Microbiome sequence datasets are typically high dimensional, with the number of species much greater than the number of samples. The consideration of pdfs limits the issue of compositionality, as well as the focus on group vs. individual statistics limits the issue of data sparsity (considering both rare species and the length of time series). Of course, this macroecological purview does not imply any strict causality inference but rather aims to set up the basis for the predictability of microbiome group features. This is also because there is no well established data or model to identify what is truly a causal effect between species, although some advancements have been made in the field of information theory such as in Lizier and Prokopenko [15] where information flow (such as the one used in our model via TE after entropy reduction) proves to assess local causality vs. information transfer via simple TE. Arguments have also been formulated about the general validity of TE to infer causality (see James et al. [62]). However, beyond these analytics centered debates, the fundamental argument should also be focusing on what kind of interaction based on data is truly inferred, what is the interaction that is wished to be inferred, and what is the modeler choice of analytics selected to represent reality [16]. All these elements of discussion would make the interpretation of results clearer, such as the distinction between inferred networks for predicting patterns vs. inferred networks claimed to represent the physics of the biological system considered. Despite sophisticated approaches to statistical transformation (such as centered log-ratio transformation that can remove the constraint of the sum of species proportions), the analysis of compositional data may remain a partially intractable problem because RSA is the information that is available. Given these findings, promising work has been done on addressing compositional data as a significant challenge to co-occurrence network inference, but the problem is still not solved [57]. However, TE is not affected by compositional data (provided enough data are given to characterize pdfs) precisely because it uses pdfs in network inference and the pdf of RSA, raw abundance, and any transformation applied to all species is the same. A problem may arise only when data are asymmetrically transformed in a way that the pdf of one or more species is altered.

The entropy/free energy patterns (or “entropy-flow patterns”) in Figure 2 do not show any strong scale invariance as for instance in Servadio and Convertino [13], likely because no pure scale-free networks are observed in the microbiome organization. In this study, we focused on the total entropy as a utility function versus the value function defined in Servadio and Convertino [13] (based on a systemic indicator) where raw values of network variables were considered rather than TEs among them. The focus on network variable interdependence (that is between species in this context) rather than nodal values (i.e., RSA for the microbiome) leads to a higher variability in network entropy patterns. Therefore, we believe that the focus should be on network function in order to better characterize networks; this is substantiated by the higher importance of species interactions (OTE) versus species independent dynamics (represented by nodal entropy), as shown in Appendix A (bottom plot). This figure shows that OTE makes up almost the whole Network Entropy ($H_{N}$) (Appendix A top plot) (see Equation (Equation 6)) so Nodal Entropy has little importance. Entropy-flow patterns are then useful for detecting scale-invariance in the functional topology of the network and for identifying MaxEnt states. Additionally the entropy-flow patterns can reveal healthy vs. unhealthy states by considering the symmetry of the entropy distribution; if symmetrical positive and negative species interactions (TEs) are found these interactions sum up to zero leading to a healthy neutral state. The asymmetry of unhealthy microbiome can certainly relate to non-neutral states created by strong stressors, as highlighted theoretically in Borile et al. [63]; these state may not allow host individuals to keep the microbiome “on a leash” [86] that causes overgrowth of abundance and multiplication of species. However, the broken symmetry can be indeed manifesting an unhealthy state. The neutral state also coincides with the critical state because of the tendency of the network toward a scale-free organization manifested by the epdf of OTE (Figure 5), higher functional distances and smaller functional degrees (Appendix A).

To assess the robustness of microbiome networks, we considered the network topology for high thresholds values of the interspecies TE. In other words, we considered as meaningful TEs, only those above a certain threshold. According to the 80/20 Pareto principle (that states that 20% of subcomponents make up at least 80% of a system’s dynamics [87]) (note that this principle works for scale-free systems), we considered only the highest 20% of TEs for the inferred networks. These Pareto high threshold networks show that the healthy group maintains the topology while changing TE; this is because healthy networks are more scale-free than unhealthy ones (see Figure 5B, that shows a scale-free like epdf of OTE), yet scale-invariance is preserved when changing the threshold defining the scale at which the network is constructed (or observed). This scale analysis is equivalent to make experiments when random nodes are removed simulating a random attack on networks [88]; thus, we can also claim the higher resilience of the healthy network for the microbiome. However, this result is expected considering the known optimality of scale-free networks [67]. The scale-free configuration enhances stability as confirmed by the calculation of the dominant eigenvalue for both the adjacency and TE matrices; the dominant eigenvalue is the smallest for the healthy group that is a signature of network stability [3].

The “non-pure” scale-free organization of the microbiome confers the ability to adapt to different externally-driven changes and to adapt vs. a more stable scale-free topology. Overall, we suggest to focus on TE and OTE as the best indicators of microbiome function (for pairs and node functional characterization), vs. any other indicator, since those are related to species interdependence. As highlighted in recent studies (see Rivett and Bell [66]) abundance determines the functional role of bacterial phylotypes in complex communities; rare and common bacteria are implicated in fundamentally different types of ecosystem functioning [66]. Such knowledge could be used, for example, to understand how bacteria modulate biogeochemical cycles, and to engineer bacterial communities to optimize desirable functional processes. Microbiome service is here identified by any microbiome diversity indicator in analogy to how services are also expressed for large scale ecosystems. Certainly, it is true that α-, β- and γ-diversity cannot be “equated” to large scale ecosystem services (i.e., the benefits that people derive from nature and how these are quantified as “natural capital”), but any diversity measure is a valuable indicator of biological function at any scale of biological organization (see, for instance, Isbell et al. [77] and Mori et al. [89]) much more than structural indicators, as shown in this paper. Therefore, there is a desired ecosystem service-function nexus that is desirable and related to healthy states (which is the benefit individuals get from having the “right” value and patterns of macroecological indicators manifesting optimal biodiversity organization). Of course, especially in microbial ecology where the identification of species is more difficult than large scale ecosystems, there are arguments about the utility and validity of different diversity metrics such as γ vs. evenness. Nonetheless, independently of this, we argue that our analyses would result in equivalent conclusions. For instance, in our case, high γ corresponds to low evenness and vice versa; thus, biodiversity patterns would reveal opposite trends but provide the same meaning because of the γ-evenness relationship.

In our microbiome data, we considered the complementary of β-diversity over time via the Jaccard Similarity Index (JSI) and we showed that JSI is higher for the healthy than the unhealthy microbiome over time. This means that the local species richness, α, tends to be more equal to previous values over time; however, this underlines the stability of α (species organization) in the healthy state. For the unhealthy microbiome, the similarity over time is lower (i.e., higher species turnover, or higher β-diversity) such as for the corals in Zaneveld et al. [74] that are evaluated over time as a function of external stressors. In other types of ecosystems, e.g., in coral ecosystems under stress, Zaneveld et al. [74] found that the true β-diversity increases over time. In macroecology, leaving aside the debates about the many definitions of species turnover, and in an entropic context the true β-diversity is the ratio between regional (γ) and local species diversity (α) [69]. This definition is in line with the general information balance equation (Equation (Equation 2)) and the more specific diversity balance equation $H_{\gamma }=H_{\alpha }+H_{\beta }$ as in Jost [69]. An increase in β is typically associated with a decrease in α as much as we observe for the healthy microbiome, and this is also associated to fluctuations of α that are smaller than those for the unhealthy microbiome. The “proportional species turnover” (i.e., where $\beta_{p}=1-\alpha /\gamma$, when considering γ partitioned into additive rather than multiplicative components) that quantifies what proportion of species diversity is not contained in an average representative sample, is also higher. This emphasizes how our results are robust independently of the peculiar definition of species diversity indicators. In ecology these quantities are typically evaluated over space and in healthy conditions 1-β has a relatively fast decay but never goes to zero; this means that heterogeneity exists but even communities far apart have species in common. Considering space in unhealthy conditions, typically the “true” β-diversity is smaller than in healthy conditions because much more homogeneity is achieved. However, heterogeneity is a good thing as shown for ecosystems at any scale of biological organization.

The higher variability of β-diversity in healthy individuals highlights the “Anna Karenina phenomenon” for human microbiomes. The principles underlying the phenomenon states that dysbiotic individuals vary more in microbial community composition than healthy individuals paralleling Leo Tolstoy’s dictum that all happy families look alike (“each unhappy family is unhappy in its own way”). The stability-unimodal pattern of diversity is concordant with current theories looking into β-diversity vs. solely α-diversity for the stability of ecosystems [73]. This is also concordant with the network entropy pattern that is unimodally stable for the healthy group. Thus, we innovatively highlight the linkage between information exchange and diversity in biological systems. Convertino et al. [39] previously found that ecosystem hotspots are those that maximize the Value of Information (of biodiversity) which coincides with those that minimize β-diversity variability over time. The multiplicity of “unhappy/unhealthy” states is reflected by the network topology that is random for the unhealthy group, which allows many more potential unhealthy microbiome combinations. We support the position of previous studies that Anna Karenina effects are a common and important response of animal microbiomes to stressors that reduce the ability of the host or its microbiome to regulate community composition. These effects may be transient and necessary to bring back the system to the healthy state.

Similar to other ecosystems, we show that scale-invariance (that is occurring for the healthy microbiome) does not arise from an underlying criticality (where fluctuations becomes bigger and bigger causing the system to tip abruptly) nor self-organization at the edge of a phase transition. Instead, it emerges from the fact that perturbations to the system exhibit a neutral drift (also relate to small extrinsic environmental changes) with respect to the endogenous spontaneous dynamics. This *neutral* dynamics, similar to the one in genetics and ecology, shows fluctuations of all sizes simultaneously that likely determine power-law distributed species diversity (as well as power-law information exchange among species). The tipping point that was observed, i.e., between healthy and unhealthy microbiome, is a second-order critical transition where exogenous fluctuations are too large to be assimilated by the system and the microbiome tips from healthy to unhealthy. This transition is evident in the shape of the pdf of microbiome function and diversity (as microbiome service) but not in the shape of microbiome structure (unless a rescaling in size is performed, for instance for the microbial network degree; see Figure 5).

The introduction of new pathogens driven by the environment can lead to the alteration of the whole ecosystem microbiome [8]. In our case study, despite the non-explicit consideration of the disturbance agent, we found a transition in IBS individuals from healthy to unhealthy states. However, this disturbance agent was considered by Durbán et al. [51] and Martí et al. [8], who worked on the original dataset. Independently of the disturbance, healthy individuals have larger gradients of speciation events and higher growth rate for γ-diversity because they produce more species (diverse of not) to guarantee necessary/basic biological function and other functions related to extreme fluctuations. Not all species need to be present all the time and that is likely the motivation for which the average $\gamma^{\prime }$ is higher for healthy and transitory individuals than unhealthy people as well the average γ is lower for healthy ones. $\gamma^{\prime }$ seems to reflect the general dynamical systems’ pattern indicated by the Heap’s law [90] that regulates the rate of diversity produced by a system. This is associated to the Taylor’s law regulating mean and fluctuations and the Zipf’s law (in our case of RSA which influence macroecological indicators). In a more ecological purview, the species–area-like relationship in Figure 6A can also emphasize the island biogeographic effect where for islands/healthy individuals γ is lower but $\gamma^{\prime }$ is higher than the mainland/unhealthy people [91] due to optimal growth (ideally not impacted by invasions). The higher γ for unhealthy individuals is likely related to invasive species for instance attributable to external sources; healthy individuals instead, have a gut flora composed by only endemic species. In a general view, Taylor’s law regulating RSA fluctuations, Zipf’s law governing RSA distribution, Heap’s law relating γ’s growth over time, and the mass-specific Kleiber’s law are all liked together by the Pareto optimal principle of self-organized design [92,93,94,95,96] that can inform about the optimality or pathology of biological systems.

The microbiome in the gut is similar to any ecosystem: no other species at all scales of biological organization can survive optimally if the microbiome is altered. The microbiome is the linkage between the fundamental genetic organization of life and the stochastic environmental dynamics; in the context of a person’s growth, it is possible to refer to those two processes as nature and nurture. The proposed information theoretic global sensitivity and uncertainty analyses (Figure 4, left plots) allow one to map the dynamics of species considering their interactions and absolute influence, and to see how these quantities vary considering their intrinsic biological variability and environmentally driven variability. One must keep in mind that these interactions are based on mutual RSA interdependence assessed by TE, so TEs might not represent the whole “true” interactions among species; however, recent evidence points to this conclusion [65,66] but there is still a lot work to be done in this area. In the healthy state, more species (fewer in number) are influencing the collective dynamics with a more organized distribution of interactions (“hierarchical” organization), while for the transitory and unhealthy state all species (higher in number) are somehow behaving equally and likely driven by external environmental stimuli (“random” organization). This organization is also reflected by network properties (Appendix A) that can be altered for the same set of species/diversity. Researchers have found that cooperation promotes ecosystem biodiversity, which in turn increases its stability without any fine tuning of species interaction strengths or of the self-interactions (i.e., neutrality) [97,98]. Even small values of TEs (close to zero) manifesting mutualistic interactions (positive) among species can stabilize the dynamics. Stability increases with the ecosystem simplicity where the latter is related to the scale-free like organization of bacteria. On the other side, too much cooperation (e.g., dictated by networks for high values of TE) promotes instability and complex random networks. It is interesting to note that this scale-free cooperation of species leads to Taylor’s laws [29,99] between mean and variance of RSA where Taylor’s exponent is different for healthy and unhealthy groups [8]. However, this reemphasizes the connection between time dynamics, network organization, and ecological patterns of diversity and RSA [31,97,100]. In particular, it has been shown that higher-order interactions (e.g., captured by $\sigma_{i}$ in our model) have a stabilizing role [100]. These higher-order interactions are all those beyond the simple pairwise interactions whose sum indeed cannot explain the whole composition and dynamics of ecosystems [101]. We show that these higher-order interactions cannot be prevalent because some species must have an independent dynamics (captured by $\mu_{i}$) otherwise instability and tendency toward disorganized unhealthy state is very likely (Figure 4).The healthy critical state is in fact characterized by an heterogenous distribution of $\sigma_{i}$ and $\mu_{i}$ for species that is optimal for the microbiome.

The definitions of detrimental and beneficial bacteria (some of them listed in Figure 4, right plots) were based on previously published papers. We incorporated this classification in Appendix A. For instance, *Lactobacillaceae* and *AcidobacteriaGp18* are beneficial, while *Neisseriaceae* and *Campylobacter aceae* are detrimental. Of course, this is just a rough categorical classification because as we emphasize in this work, for a bacteria being detrimental or not is a function of relative abundance and network topology rather than just being present or not in the microbiome or other independent properties without considering the bacteria collectivity. Microbiome functional network topology defines how all bacteria behave synergistically and that synergy brings a healthy or an unhealthy state. Additionally, the functional topology characterization, for instance determined by OTE, can avoid the issue of determining precisely what true “species” are that is a debated topic in microbial ecology. The focus is on portfolios of interacting species whose interaction is responsible for the microbiome dynamics/state. This result sheds some light into a vision where a diminishing role of network hubs (considering total information flow) is reported as found by other studies [102]. The least relatively abundant species for the unhealthy microbiome are the most interactive and the least detrimental. On the contrary, the most relatively abundant species (Appendix A) for the unhealthy microbiome are the least interactive and the most detrimental. These analyses considering the activity of species show the importance of weak ties (interactions) for the healthy and unhealthy groups. This is in accordance to general dynamical principles such as the Granovetter principle about the strength of weak ties for the systemic dynamics of a complex system [103]. For the healthy microbiome, the highest RSA species interact the least and these species are the most beneficial. These species–specific analyses, when verified, are useful for detecting species that are more beneficial or detrimental and this knowledge can lead to design probiotic treatment, microbiome transplants [104], and large scale ecosystem microbiome controls [105] for instance.

Universality in human microbiota dynamics [82], whether present, can be ideally manipulated in a similar or even identical fashion in multiple individuals for population health. Following the discovery of universality and the demonstration of beneficiary effects of specific interventions, microbiome engineering efforts can be applied to a large number of people. In this way, microbiome engineering will be highly cost-effective as a public-health based approach. This is in sharp contrast to the excessive cost of “precision-medicine” approaches that try to target individual microbiome dynamics by considering it as a purely individual-based feature. Current frontier topics are also related to the understanding of how the microbiome and functional brain networks “communicate” [106]. It seems that the nervous system contribute to dictate which microbes inhabit the gut; this in turns affects emotional response and long term well being beyond short-term health.

The hypothalamic–pituitary–adrenal axis (HPA axis) is a primary mechanism by which the brain can communicate with the gut to help control digestion through the action of hormones [106]. It seems that the nervous system, through its ability to affect gut transit time and mucus secretion, can help dictate which microbes inhabit the gut, which in turns affects emotional response and long-term well being beyond short-term health.

## 5. Conclusions

An information theoretic model for the inference of microbiome networks and the related biodiversity organization over time is proposed. The model consists in the assessment of transfer entropy-based species interactions after entropy reduction calculations that remove the second-order indirect interactions between species as in the works of Lizier and Prokopenko [15] and Lizier [56]. Maximum entropy networks are then extracted considering the highest information content without model overfit; overfitting is avoided by removing the redundant variables for the simplest MENet, that is an Optimal Information Network. Species interactions should be interpreted in terms of species predictability rather than causal mechanisms due to the data- and model based-dependence of the inferred interactions [62]. The macroecological validation of the model was performed considering the ability to simultaneously predict the pdf of α-diversity, γ-diversity growth, species similarity (1−β) decay, and the RSA-rank profile. This validation allowed predicting other biodiversity patterns such as the Preston’s plot of average species richness dependent on species RSA. Considering the application of the model to healthy and IBS symptomatic individuals, the following points are worth mentioning without lack of generality.
Directed species interdependencies and phase transitions of the microbiome over time were detected. The healthy microbiome is characterized by balanced positive and negative species interactions vs. the unhealthy microbiome where most species interactions are positive. The balanced interactions were evidenced by the symmetrical pattern of the total network entropy as a function of the pairwise information flow (TE) vs. the positively biased asymmetrical pattern of the dysbiotic microbiome. The healthy symmetrical network entropy pattern underlines the neutral “sum to zero” dynamics of species interactions (based on RSA); the same neutrality was found for biodiversity of large scale ecosystems at stationarity that are driven predominantly by intrinsic ecological stochasticity (ecological drift). On the contrary, unhealthy microbiome entropic patterns are affected by environmental disturbances; the positive bias in information flow (that may relate to infections and antibiotics, as shown in the original data [51]) causes an overgrowth in RSA of many opportunistic species as well as the generation of new detrimental species. The categorization of beneficial and detrimental species was based on published literature; however, we emphasize how important it is to consider collective bacteria topology vs. individual bacteria behavior when defining health and disease;The healthy state is characterized by the highest total species diversity growth rate $\gamma^{\prime }$ (leaving aside the transitory microbiome) and the lowest loss of species similarity over time, i.e., species turnover (${\left(1-\beta \right)}^{\prime }$). A relationship similar to the species–area relationship for large scale ecosystems [70] was found between γ-diversity and the number of species generations with an exponent equal to 0.20 on average. The fact that the healthy microbiome has the lowest average total diversity (γ) is in contrast to what is observed in large-scale ecosystems at stationarity where the highest total diversity correspond to the stable and supposedly healthy state [78]. However, we speculate that an optimal diversity growth is oriented toward maximizing growth rate rather than total diversity (as according to many Pareto portfolio theories). The latter can lead to over-redundancy of microbial interactions and instability as observed for the dysbiotic microbiome; the highest γ diversity for unhealthy ecosystems is related to non-endemic species. Hence, we tend to challenge the diversity–health–stability hypothesis when for diversity the total systemic diversity γ is solely considered without the consideration of “invasive” species and $\gamma^{\prime }$;We observed a phase transition of the second order from the healthy to the unhealthy state and vice versa. The transition from healthy to unhealthy is characterized by typical signs of transitions observed in many complex systems [107], i.e., an increase and a decrease in mean and variance of species diversity while approaching the transition (“critical slowing down”). In the unhealthy state the variance of α is higher than in the healthy state and concentrated around two values which underline the likely chaotic-like dynamics of the microbiome. In terms of microbiome functional network topology, a transition between the scale-free to the random network topology is observed. The critical state, defined by a scale-free-like organization of microbial species interactions, coincides with the neutral state (i.e., for the symmetrical network entropy pattern) emphasizing how criticality does not necessarily occur at critical phase transitions, particularly for second-order transitions as in this case. Rather, criticality can coincide with neutrality in open energy dissipative systems, as observed in other complex systems [20]. Criticality at the phase transition can favor gut adaptability but may pose high risks to tip to unhealthy states. Neutrality implies lower topological complexity and higher dynamical stability (corresponding to higher symmetry, higher organized information exchange, lower entropy/total information, higher diversity, and higher predictability (or information content)) considering the scale-free and small-world functional and structural organization of the microbial network. We emphasize how the healthy local stable state is dynamically flexible because of the lower entropy (i.e., higher free energy) and more predictable due to the more organized collective behavior of species; however, due to the gradient in entropy moving from locally stable unhealthy conditions to the globally healthy stable one is hard;A probabilistic linkage was found between microbiome function and services, defined by species interaction topology and biodiversity organization, respectively. We did not find any correspondence between microbiome structure and function, which emphasizes the non-linearity between the two and the importance of assessing function rather than structure in biological networks. We propose the total Outgoing Transfer Entropy (OTE) as the measure to identify the most influential nodes (and pairs); these nodes are able to predict the behavior of all other connected nodes, as well as of the whole microbiome. OTE is largely determining the total entropy of the network compared to the sum of nodal entropies whose contribution is negligible. This emphasizes even more the role of collective behavior vs. individual nodes considered in isolation. The highest OTE nodes have the lowest RSA, and these are the most beneficial and the most detrimental bacteria for the dysbiotic and healthy microbiome. A scaling law was found between OTE and RSA with an exponent close to 1/4 that is similar to the mass-specific Kleiber’s law [81] where the species specific metabolic rate is the OTE and the mass is the RSA. A power-law distribution for the microbiome function (i.e., the sum of nodal OTE) was found for the healthy state (with an exponent ∼2 that implies finite mean but infinite variance suggesting how the healthy condition is prone to perturbations enhancing fluctuations of all sizes) despite no information (or resolution) invariance being detected in the network entropy pattern (see Servadio and Convertino [13]). The lack of scale invariance in the entropy/free-energy phase space may imply the metastability of the microbiome that can indicate its resilience in terms of ability to move quickly from one state to another.

## Figures and Tables

**Figure 1 entropy-21-00506-f001:**
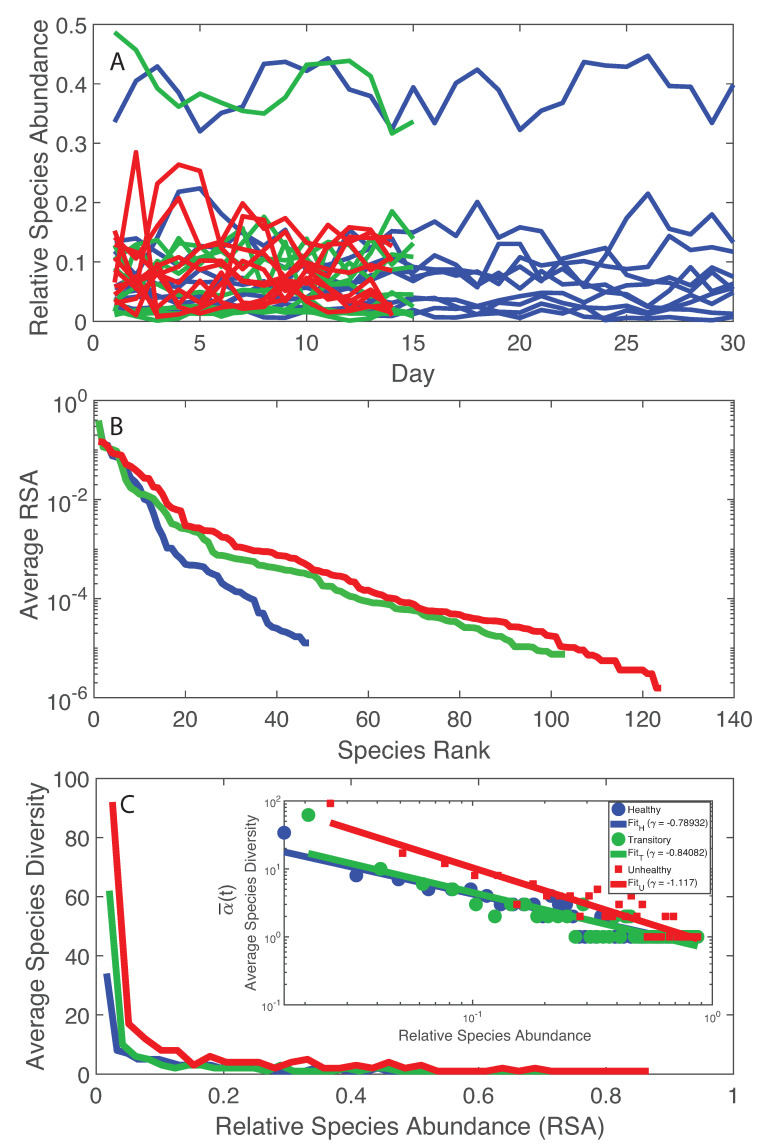
RSA trajectories, RSA-rank, and Relative Species Abundance. Blue, green and red curves refer to the healthy, transitory and unhealthy microbiome, respectively. **A**: RSA time series for all individuals before LCM; **B**: average RSA-rank pattern; and **C**: average species diversity vs. RSA (the inset shows the same pattern in a loglog scale. The healthy microbiome shows smaller fluctuations in species diversity α vs. RSA and one regime when considering the RSA-rank profile. An inverse scaling law was detected between the average species diversity and RSA (inset in **C**).

**Figure 2 entropy-21-00506-f002:**
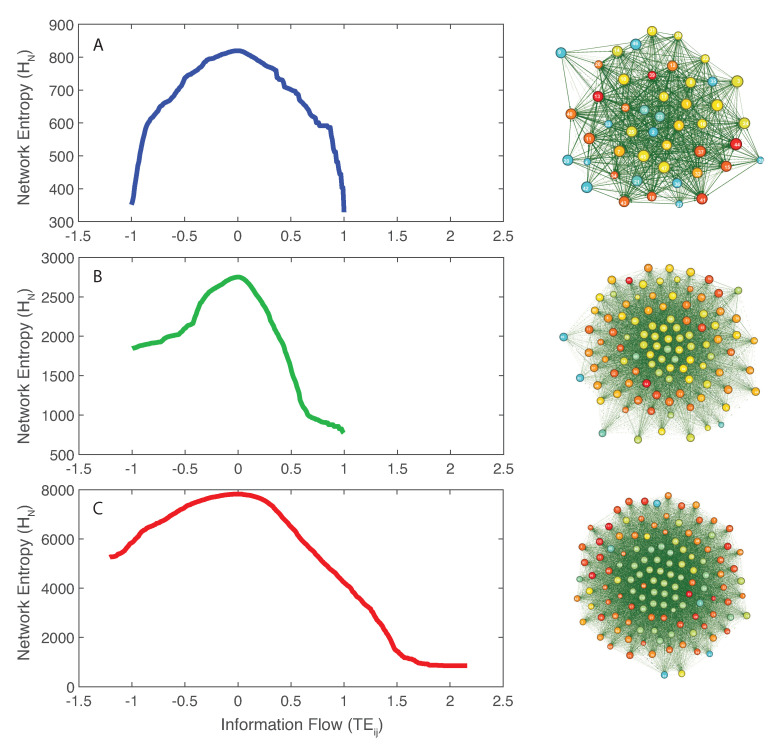
Network entropy patterns and inferred Optimal Microbiome Networks. Network entropy dependent on the pairwise information flow (TE) (**left** patterns) and extracted Optimal Information Networks for the microbiome on the **right** (Maximum Entropy Networks after node redundancy exclusion). **A**, **B**, and **C**: network entropy patterns for the healthy, transitory and unhealthy microbiome. The size of each node is proportional to the Shannon Entropy of the species; the color of the node is proportional to the structural degree (in Appendix A, the color of each node is proportional to the sum of total outgoing TEs of each node (OTE); the higher is the OTE, the warmer is the color); the distance is proportional to exp(−MI(X,Y)) where MI(X,Y) is the mutual information between species RSA *x* and *y*; the width of each edge is proportional to the pairwise Transfer Entropy; and the direction is related to TE(i−>j); the direction of this edge is from *i* to *j*.

**Figure 3 entropy-21-00506-f003:**
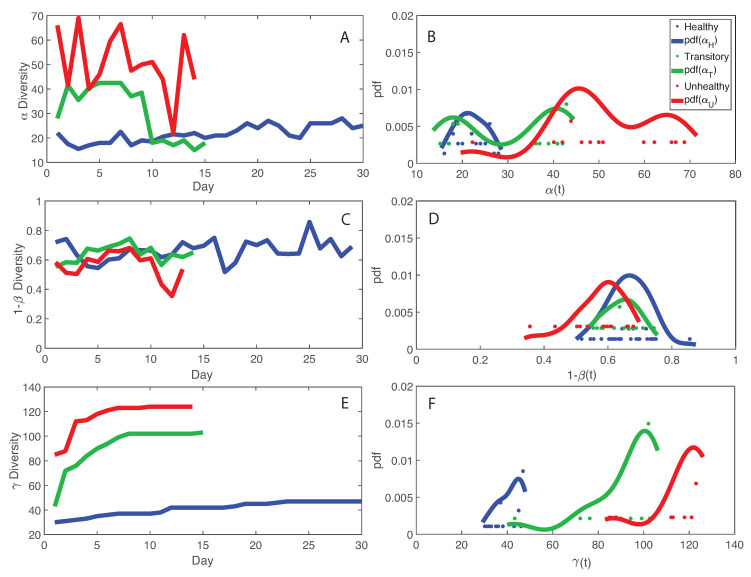
Macroecological indicators of microbiome networks and probabilistic characterization. Average α, species similarity 1−β, and total diversity γ are plotted as a function of time. Their probability distribution is shown on the right. **A**, **C**, and **E**: α, 1−β, and γ diversity over time. **B**, **D**, and **F**: pdf of α, 1−β, and γ diversity.

**Figure 4 entropy-21-00506-f004:**
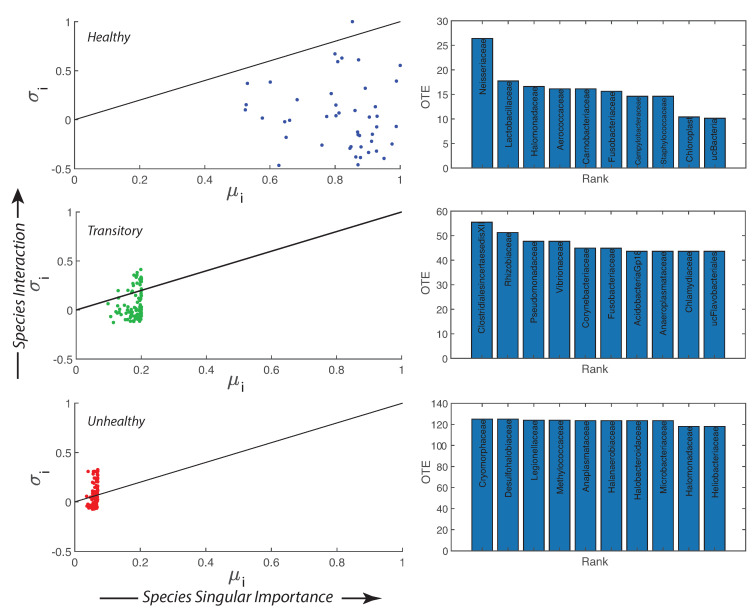
Importance and interaction of microbial species, and top 10 most active species species. Transfer Entropy Indices: σ is describing species interaction and is calculated as the ratio between the total Outgoing Information Flow (OTE) ($OTE\left(j\right)=\sum_{i}TE_{j\to i}$) and the Total Network Entropy, while μ is describing the species importance as the ratio between the Nodal Entropy (Shannon Entropy) and the Total Network Entropy. The continuous line in each σ-μ plot (**left**) shows the critical edge that describes a state between regularity and chaos. On the **right** plots, the top 10 most active species in terms of OTE (and least relatively abundant) are ranked for the healthy, transitory and unhealthy microbiome (from top to bottom). These species are the most detrimental for the healthy group and the most beneficial for the unhealthy one.

**Figure 5 entropy-21-00506-f005:**
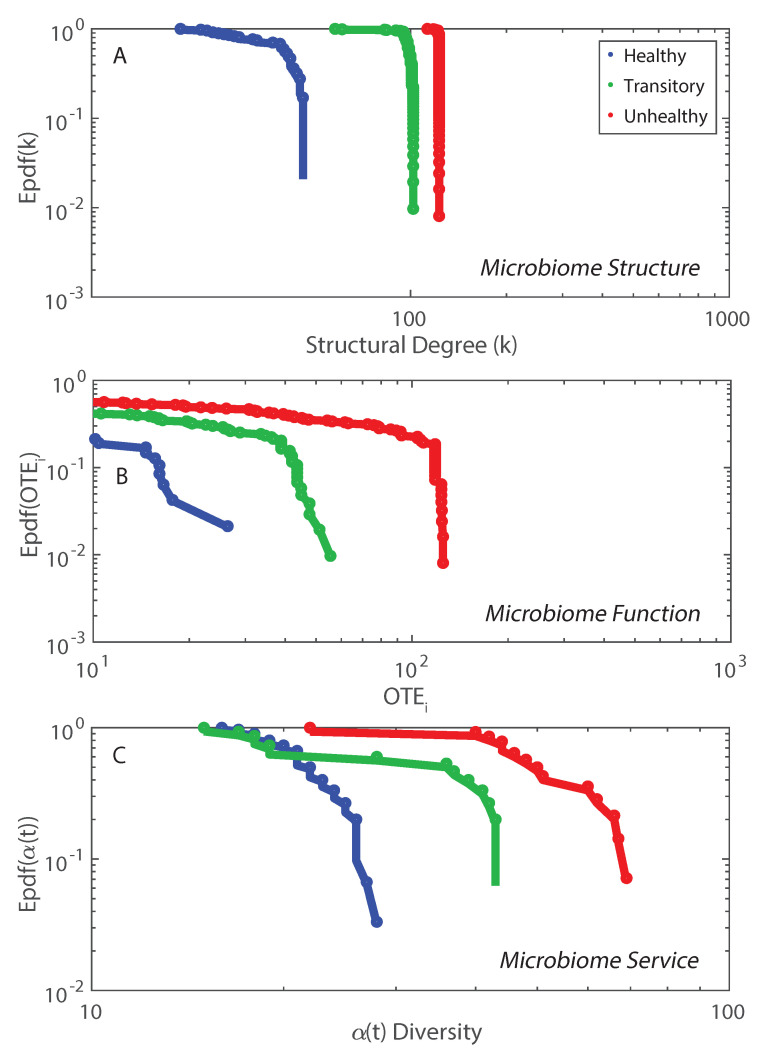
Exceedance probability distribution of microbiome structure, function, and service. Network degree, total outgoing transfer entropy (OTE) of each node, and α-diversity over time characterize the structure, function and service of the microbiome network (**A**, **B** and **C** plots).

**Figure 6 entropy-21-00506-f006:**
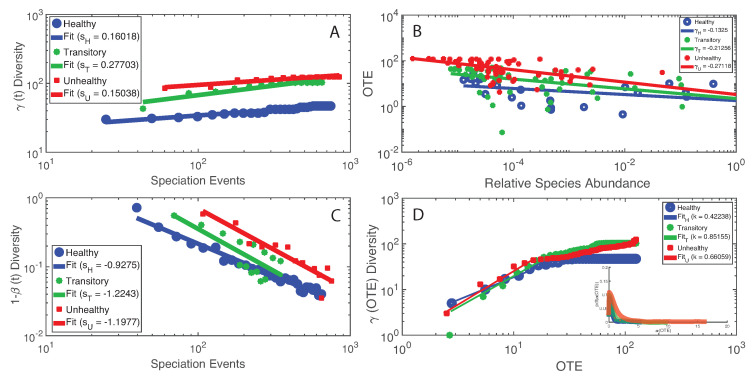
Macroecological scaling patterns and predicted species interactions. (**Left**) The scaling of total γ-diversity and species similarity 1−β dependent on the number of speciation events (**A** and **C**) that is the number of new and existing species introduced until the time considered; speciation time is a proxy of the sampling area over time. (**Right**) The scaling of OTE vs. RSA (**B**) and γ-diversity vs. OTE (**D**) that consider the mutual variability of information exchange and macroecological indicators of the microbiome.

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
