# Peer review of "Optimal Microbiome Networks: Macroecology and Criticality"

_entropy, 2019, doi:10.3390/e21050506_

Round 1

Reviewer 1 Report

First, I would like to state that the authors addressed most of my previous concerns and the manuscript is now much more understandable from a microbiologist point of view. However, while the clarity has been improved, there are still problems with the language that sometimes make interpretation of the meaning difficult. The manuscript should be revised by a translator.

Discussion is still mixed in with the results, but it helps the story, so maybe the results and discussion should be merged under the same section.

I have a new concern with the time series reconstruction presented. While it clarifies the discrepancy in the time lengths presented in the previous version, it reveals that the method used to compensate for different sampling time lengths infers a lot of data. I find it concerning that there is an obvious bias in the data available, healthy > transitory > unhealthy and I was afraid that this bias could be driving the results presented. However, the authors say in their discussion that it does not change the patterns detected. Therefore, the approach may be legitimate, but as an experimentalist, the practice of extrapolating data does not sit well with me. Why not analyse the data for the period of time where all the data is available if the results are similar?

Author Response

Reviewer 1

First, I would like to state that the authors addressed most of my previous concerns and the manuscript is now much more understandable from a microbiologist point of view. However, while the clarity has been improved, there are still problems with the language that sometimes make interpretation of the meaning difficult. The manuscript should be revised by a translator.

We thank the reviewer for this comment. We went through the whole manuscript again and adjust several mistakes or weird expressions. We believe the current state is well satisfactory at this stage. 

Discussion is still mixed in with the results, but it helps the story, so maybe the results and discussion should be merged under the same section.

We thank the reviewer for this comment. We took care of removing the majority of extra discussions present in the Results section and bring those into the Discussion section without compromising the presented story.

I have a new concern with the time series reconstruction presented. While it clarifies the discrepancy in the time lengths presented in the previous version, it reveals that the method used to compensate for different sampling time lengths infers a lot of data. I find it concerning that there is an obvious bias in the data available, healthy > transitory > unhealthy and I was afraid that this bias could be driving the results presented. However, the authors say in their discussion that it does not change the patterns detected. Therefore, the approach may be legitimate, but as an experimentalist, the practice of extrapolating data does not sit well with me. Why not analyse the data for the period of time where all the data is available if the results are similar?

The datasets have 3 health groups which are respectively comprised of 2 healthy individuals (A and B), 2 transitory individuals (C and C1) and 2 patients (P1 and P2) with the length of 30 days for A, 15 days for B, 15 days for C, 9 days for C1, 9 days for P1, 14 days for P2. In order to get the group data from datasets of 2 individuals with different lengths, we put them together by using the method of Least Common Multiple (LCM) that does not destroy the wholeness of each time series, but just replicates them to make the length of the time series identical. Such time-series reconstruction does not lead to any changes in their probability distribution function (pdf) that is what is needed to infer the potential species interaction network. With longer time series the fit of the pdf is generally speaking better than with shorter time series.

The focus we studied in this work is to predict selected biodiversity patterns by analyzing statistical features of observations and inferring networks for the microbial ecosystems. Therefore, the time-series reconstruction does not change the patterns detected. The model we used to calculate TE of two random variables requires that the length of one time-series variable match that of other one; this is also why we decided to increase the length of the time series. We explained all these technicalities in a revised paragraph of the subsection ‘’Time Series Reconstruction’’.

Reviewer 2 Report

The manuscript describes an information theoretic framework to build directed microbe-microbe interaction networks based on time series data. Even though I would not consider myself an export in information theory the presented methodology seems sound and warrants the presented results. The manuscript seems to be a revision to a previous evaluation (red sections in the manuscript) and I could not detect any major drawbacks. Thus, I will only provide some recommendations in order to improve the manuscript.

The inference of interactions in the microbial communities based on sequencing data is often plagued by the inherent compositionality of this data type (https://doi.org/10.3389/fmicb.2017.02224). This can often lead to spurious associations and is not immediately clear to me if the proposed methodology avoids that caveat. It would be worthwhile if the authors could include a brief comment on how to deal with compositional data in their framework. For instance would their method still work when apllying a log-ration transform such as clr?  

As the authors specifically discuss causality, directionality and transfer entropy it would be interesting to see which additional information is gained from that. For instance if the network were constructed with non-directional measures such as usual mutual information would the main conclusions be lost? I don't consider this essential for publication though.

I applaud the authors for making the source code public. Some documentation or usage instructions might help in making this more accessible. This is also not essential for publication.

Author Response

Reviewer 2

The manuscript describes an information theoretic framework to build directed microbe-microbe interaction networks based on time series data. Even though I would not consider myself an export in information theory the presented methodology seems sound and warrants the presented results. The manuscript seems to be a revision to a previous evaluation (red sections in the manuscript) and I could not detect any major drawbacks. Thus, I will only provide some recommendations in order to improve the manuscript.

The inference of interactions in the microbial communities based on sequencing data is often plagued by the inherent compositionality of this data type (https://doi.org/10.3389/fmicb.2017.02224). This can often lead to spurious associations and is not immediately clear to me if the proposed methodology avoids that caveat. It would be worthwhile if the authors could include a brief comment on how to deal with compositional data in their framework. For instance, would their method still work when applying a log-ration transform such as clr?

We thank the reviewer for providing additional comments on the issue of compositionality. However, as largely emphasized in our first rebuttal letter we actually provided already an extended comment about the issue of compositionality (see Lines highlighted in green in the Discussion section).

Compositional data are measures providing the information of proportions, percentages, etc. The sum of the compositional data is a constant value, which is a constraint for applying these data to some statistical analyses such as PCA. Centered log-ratio (clr) is one of the most common transformations that can remove the constraint. In clr, sample vectors undergo a transformation based on the logarithm of the ratio between the individual elements and the geometric mean of the vector. It removes the constraint by using a log calculator that just numerically changes the scale of data, but does not change the statistical features. The computation of variables in information theory including TE is based on probabilities. Thus, we have reason to believe that the proposed TE-based model can be well applied to the data after clr transformations (of course); the results from these transformed data are extremely likely to be similar to the resulting patterns found in this study because those results would be based on a transformation of values that do not change the shape of pdfs of species.

As the authors specifically discuss causality, directionality and transfer entropy it would be interesting to see which additional information is gained from that. For instance, if the network were constructed with non-directional measures such as usual mutual information would the main conclusions be lost? I don't consider this essential for publication though.

We thank the reviewer for the comment. However, we point out that we indeed calculate already MI to establish whether or not there is an interaction and after we calculate TE to establish the direction of that interaction. Thus, we provide more information than what MI networks would provide. Additionally, we should say that the topology of the network is not impacted by MI nor TE because topology (at least structurally) is about organization of node connections and that is not dependent on the direction of these connections.

Considering the causality or interactions themselves between 2 different variables, the effect of variable A on B ought to be different from the effect of B on A. Let’s consider an oceanic fish community as an example, where A denotes a fish species and B stands for one of the environmental factors such as sea surface temperature. Naturally sea surface temperature significantly affects the physiology and behavior of this fish species, but not vice versa. It is therefore necessary to study the causality with its directionality and this is what we did in our paper.

As the result of using the directional transfer entropy model, we can infer directed networks that bring more informative structures and functions for modeling the ecosystem than undirected networks. Additionally, we can use the important concept of OTE for describing the node activity in a collective network sense. If we use symmetrical (non-directional) measures such as mutual information, we just can obtain an incomplete information from the MI-inferred undirected network. Thus, we believe that TE and OTE are important and necessary metrics to consider.

I applaud the authors for making the source code public. Some documentation or usage instructions might help in making this more accessible. This is also not essential for publication.

We thank the reviewer for this. We will supplement more annotations for the source codes and create a webpage with documentation for the usage of the model for microbiome ecosystems as well as for other ecosystems.

This manuscript is a resubmission of an earlier submission. The following is a list of the peer review reports and author responses from that submission.

Round 1

Reviewer 1 Report

The manuscript presents a case study analysis applying theoretical concepts to human gut microbiota data. While the combination is highly interesting and the work has merit, the manuscript has to be improved in several ways if it is to appeal to scientists working on the gut microbiota. In its current form, it is not sufficiently comprehensible, and misinterpretation is highly likely. The opinions I provide come from a microbiologist/systems biologist, therefore I am not able to comment on the validity of the physical concepts and equations presented. These should be reviewed by an expert in this field before this work is published.

My greatest concern is that building a theoretical model to predict patterns based on n=2 individuals per group seem very unreliable. Given the nature of the data, I would like to see a validation of some kind on another dataset.

The methods refer to another paper as a source for the data, but it is not enough information. The paper cited has a larger dataset, and it is unclear which samples were used. The referred paper is already a meta-analysis, therefore is it unknown if the data for each group even comes from the same method.

The authors say samples with 30 days of data were used, but only the healthy group appears to have 30 days in the figures, while the others are only 15 days. Could this have biased the results? Why not include only 15 days for each group?

The authors present the data as the “abundance of species”. There are two problems there. First, in the referenced paper for the data source, the data is presented as relative abundance. Why are they presenting abundance data and not relative abundance? It is critical for interpretation to understand what the data actually represents. Relative abundance data is not the same as absolute abundance data and it should be clearly stated what is what. The assumptions derived from the data and the mathematical interpretation are not the same. If this is irrelevant for your model (which I doubt), it should be clearly stated how it was taken into account. Anyhow, the data presented should include units of measurement. Second, the term “species” is not well defined. With this type of data, the analysis is usually presented as operational taxonomic units (OTUs) or a specific phylogenetically defined level (class, order, family, etc..) , which are not necessarily the species level. From figure 4, the family level appears to have been used. This should be clarified.

Many terms also have an unclear meaning (e.g. number of unique species) and some could be confused with their biological interpretations: e.g. microbiome function and speciation events.

There are no figure legends in the PDF, the figures are sometimes not referenced in the text (e.g. Figure 1A), and there is an extra overview figure. The figures should be identified by a panel (a,b,c, etc.) not top, or left side.

There are multiple repetitions in the text, the ideas presented twice. For example, in the abstract on lines 8 and 20 and lines 280–282.

The structure of the text is also confusing:

There is too much discussion in the abstract.

The result section begins very abruptly given that the methods are presented afterwards.

Some elements of the result second should be in the figure legends: e.g. lines 151 and 154

There are discussion elements in the result section (many citations).

The conclusions are after the methods.

There are discussion elements in the methods: lines 454–455

There are parts that are not relevant to the work presented:

Lines 28–31 and lines 357–362

There are statements that are not supported by citations or results: when the authors mention detrimental or beneficial species. On which basis are they classified as such?

Some acronyms are not defined in the text when they are first used (e.g. SW)

There are several language mistakes on lines:

41, 72–73, 103 (2x), 111, 117, 156–157, 176, 181, 188, 190, 209, 210, 238, 268, 269, 300, 301, 323, 325, 348, 386, 391, 479, 488, 525, 542, 554, 564, 596, 617–618

Reviewer 2 Report

Summary

In this manuscript, the authors propose an information-theoretical approach for microbial network inference. They then compare microbial networks inferred from healthy and from sick (IBS) individuals and conclude that healthy networks are small-world and hierarchically organized, whereas unhealthy ones are random and have a skewed proportion of positive to negative edges. While this manuscript contains some interesting ideas, the analyses suffer from a number of major issues, as detailed below. 

Major

The greatest concern regarding this study is the data used to draw conclusions. It is not clear which data sets precisely were used to build the “healthy”, “unhealthy” and “transitory” networks. The “unhealthy” network was presumably built from metagenomics data generated by Durban et al. for IBS subjects (https://www.ncbi.nlm.nih.gov/pubmed/23889283) whereas the “healthy network” was probably built from data generated by Caporaso et al. (https://genomebiology.biomedcentral.com/articles/10.1186/gb-2011-12-5-r50) and/or David et al. (https://genomebiology.biomedcentral.com/articles/10.1186/gb-2014-15-7-r89), since they both consist of long time series. In this case, they represent 16S data. Thus, it is unclear whether differences between the unhealthy and healthy networks are due to the sequencing technology employed (metagenomics versus 16S) or due to differences in the biology. This problem of confounding factors questions the entire work of the authors.

If only one network is available per group (here healthy/unhealthy gut microbiota), it is not possible to make general statements on topological differences, since statistical significance cannot be assessed. In other words, it is unclear whether the networks are good representatives of their respective group. 

The authors claim that the health-associated network is scale-free, whereas the dysbiosis-associated network is more similar to a random network. Such statements should be quantified with appropriate statistical tests, e.g. the goodness of fit of a power law to the node degree distribution.

It is also unclear how exactly the networks were built. How was the data from multiple individuals combined to generate one network per health status? How does the authors’ method address compositionality, a common issue in microbial network inference (e.g. discussed in https://www.ncbi.nlm.nih.gov/pubmed/26905627)? How did the authors treat rare taxa, which are absent in a large number of samples?

Finally, the authors do not attempt to validate their network inference approach in any way. The “validation” with probability distributions of macro-ecological properties is insufficient, since these involve only a few parameters, which can be obtained in a number of ways (overfitting). A comparison to existing approaches on simulated data or a literature-annotated network would allow readers to better judge whether this framework produces correct results.  

Minor

The manuscript is not well written. The structure has to be improved to clarify what the main findings are and how they are derived from the data.

There are no figure captions, which makes it hard to understand the figures.

7: neutral symmetrical pattern

It is misleading to apply the word “neutral” to interaction patterns when it is also used in the same paper to refer to community dynamics. The word “balanced” fits better with respect to interaction signs.

38-39: These shifts are typically associated with the gut that is the most diverse part of the human body considering the bacteria holobiont [11,45].
Ref 11 does not contain experimental work validating the statements made in 36-39. 

50-51: For instance the same level of diversity can be achieved via different network topologies that may lead to different health states [6].
Although Ref 6 is an example of the microbiome affecting health status, this paper does not address how microbial diversity is generated through different network topologies, or how network topologies lead to different health states.

58-61: however, nobody demonstrated how the microbiome network is different for these healthy and unhealthy groups (i.e. ”states” generally speaking when not focused on a particular subpopulation) and how the transition from one to another occurs.
Healthy and unhealthy states of human-associated microbiota have been compared on the network level before. For example, the reference below compares differences in topology and modularity between IBD and health-associated networks: 
 Baldassano, S. N., & Bassett, D. S. (2016). Topological distortion and reorganized modular structure of gut microbial co-occurrence networks in inflammatory bowel disease. Scientific reports, 6, 26087.

67: where the number of connections is n(− 1)/2, because of the directed topology of the network 

A directed network has (n-1)*n connections without counting self-interactions. The indicated number of connections is valid for undirected networks. 

70: A variety of different models have been proposed to infer network structures from small and large datasets. 

Here, the authors should cite and summarize the existing literature on microbial network inference. 

94: From neutral to niche states a critical transition is typically observed where species network organization exhibits scale-free behavior [9,13,14,26,31].

Except for ref 26, the indicated literature does not discuss critical transitions. Ref 26 does discuss critical transitions, but not from neutral to niche-driven dynamics. I also do not think that it is accepted knowledge that neutral to niche transitions are accompanied by a change to a scale-free network structure.

114: By a simple cursory analysis it is evident that the average abundance of the healthy microbiome is lower than the average abundance of the unhealthy microbiome independently of the species; however, the maximum abundance is higher for the healthy microbiome and that species is one of the the most beneficial for health.
In general, the authors used data sets with relative abundances, so no statement on differences in total abundance can be made. A recent data set with absolute abundances obtained through flow cytometry suggests that healthy gut microbiota have higher total abundances than diseased ones (https://www.nature.com/articles/nature24460).

If the authors meant abundance differences between specific taxa, where is the data or figure demonstrating this in the manuscript? Which prevalence threshold was used, and was the data rarefied to equal depth? Figure S1 does not show differences in species diversity between the health states. 

125: Figure 1C shows that the decay in richness over abundance is higher for the unhealthy microbiome; this result underlines the fact that higher diversity does not imply stability because of the suboptimal, yet unsustainable distribution of species in the unhealthy microbiome.
How can we derive from the species distribution that it is sub-optimal and unsustainable? How is an optimal species distribution defined?

153: Please define TE in more detail before referring to it in the manuscript. 

157: The transition in network topology, from random to small-world (tending toward a scale-free network) for the unhealthy and healthy groups, is manifested also by the shift in total entropy pattern (left plot in Fig. 2).
What criteria were used to define a switch from random to small-world topology? Can the authors compare their inferred networks to appropriate null models to define these switches? 

169: Why do the plots in Figure S7 only show a subset of nodes, and are these different for high and intermediate TE? The statement that the unhealthy microbiome has more positive interactions only holds for the top 10 TE. Does this relationship hold when all nodes are considered? 

190: Species collective interaction and singular importance is show in Figure 4 by plotting the information theoretic global sensitivity indices σi and µi (see Methods).
I could not find the definition of these indices in the methods. 

195: as found by other studies 

Which studies? They need to be cited.

196: The least abundant species for the unhealthy microbiome are the most interactive and the least detrimental.
How is “detrimental” defined in this context? Is it through the sensitivity indices defined above? I am not sure how these sensitivities relate to a more biological interpretation of their negative effect. There are no health indicators used in the manuscript that would demonstrate the detrimental effect of these species. 

204: The top 10 TE species are the most dangerous bacteria (”antibiotic”) but their abundance is small for the healthy microbiome; this means that these bacteria are controlled by all other good bacteria. 
What is the biological basis for describing bacteria as dangerous and “antibiotic”? 

208: The epdfs show how microbiome function is much more suited to show network topology versus microbiome structure.
Microbiome function implies function in the biological sense (e.g. metabolic capacity). In this case, the authors refer to it as a topological property, which is misleading. In addition, it is unclear what is meant by “microbial service”. If used in the sense of “ecosystem services”, it is also misleading. Microbial communities usually have a high functional redundancy, thus γ diversity cannot be equated with ecosystem service.

239: The model is used to infer a microbial network suitable for predicting selected biodiversity patterns characterizing space-time organization of α, β,and γ diversity. Thus, the purpose of the model is not to infer causal (or ”true”) species-species interactions among bacteria.
If this is not the goal of the model, then why do the authors refer to the top TE bacteria and claim they find “dangerous” bacteria? 

249: The lower entropy in species collective interaction has certainly implications for data collection, potentially implying less data needed for characterizing healthy microbiomes.
This study is based on two analyzed healthy microbiomes. I am not sure whether the limitations of the data warrant such general statements at all. 

259-262: However, we believe that the focus should be on network function in order to better characterize networks; this is substantiated by the higher importance of species interactions versus species independent dynamics as shown in Fig. S5.
Please clarify how Figure S5 demonstrates that species interactions are more important than species-independent dynamics. 

323: In our case study, despite the non explicit consideration of the disturbance agent, we see a transition in IBS individuals from healthy to unhealthy states.
Which transition? As far as I am aware, the authors studied two healthy individuals and two IBS individuals, but no transition from one state to the other within the same individual. 

324: The microbiome is like the gut of any ecosystem: no other species at all scales of biological organization can survive optimally if the microbiome is altered. The microbiome is the linkage between the fundamental genetic organization of life and the stochastic environmental dynamics; in the context of a person’s growth it is possible to refer to those two processes as nature and nurture.
The gut of any ecosystem? Environmental dynamics does not need to be stochastic. In general, the meaning here is not clear.

328: The proposed information theoretic global sensitivity and uncertainty analyses (Figure 4) allows one to map the dynamics of species considering their interactions and absolute influence, and to see how these quantities vary considering their intrinsic biological variability and external variability.
Earlier in this discussion the authors state that their methodology is not appropriate for inferring causal species-species interactions, so why is this stated here? 

347: We show that these higher-order interactions cannot be prevalent because some species must have an independent dynamics (captured by µi) otherwise instability and tendency toward disorganized unhealthy state is very likely (Fig. 4).
This is a strong claim and I cannot interpret Figure 4 in a way that allows me to reach a similar conclusion. This is the first section in the manuscript where the authors address higher-order interactions at all. I do not see where the authors conclude that some species must have independent dynamics, or how the µ statistic captures this independency.

350: Universality in human microbiota dynamics can be ideally manipulated in a similar or even identical fashion in multiple individuals.
As with the higher-order interactions, I do not see how the results support the claim that microbial dynamics are universal. In addition, the paper claiming universality of microbial community dynamics in the gut is not cited (https://www.nature.com/articles/nature18301).

361: It seems that the nervous system, through its ability to affect gut transit time and mucus secretion, can help dictate which microbes inhabit the gut; this in turns affect emotional response and long term well being beyond short-term health. 

What is the link with the analysis done in this manuscript?

366: Martí et al. used already published data. It would be collegial to credit the original authors who generated the data. Moreover, it is not obvious why these six individuals were selected, when the Martí publication included data on 99 individuals. Finally, the authors do not describe how they preprocessed the data. 

408: It is not clear whether the authors use absolute or relative abundance, and whether the TE is large for species that are below the detection limit in the same samples. It would be good to know whether the TE is robust to compositionality and sparsity.  

549: The rationale for considering the shortest paths is related to the exponentially large ensemble of distances as a function of the number of nodes and the fact that biological systems always optimize information transmission.
Unless the authors can back up this claim with references, it is better to report this statement as an assumption rather than a fact. 

565: The model consists in the assessment of transfer entropy based species interactions after entropy reduction calculations that remove the spurious direct interactions related to indirect interactions between species. 
It is not clear whether this calculation indeed removes indirect interactions between species. The authors did not conduct a single test to check that this is the case.

582: On the contrary, unhealthy microbiome entropic patterns are affected by environmental disturbances; the positive bias in information flow (e.g. related to infections and antibiotics) causes an overgrowth in abundance of many opportunistic species as well as the generation of new detrimental species.
There is no biological interpretation of the identified species; there is no basis for the claim that opportunistic and detrimental species increase in abundance. The authors only analyzed TE and no relevant biological properties. 

The authors should make their code freely available, e.g. on Github. The code and its usage should be well documented.

There were many grammatical errors, some of which are listed below. The manuscript has to be carefully checked for further errors of this type.

L. 41 it not yet settled upon -> it is not yet settled upon

L. 130 a result that likely confirm -> confirms

L. 316: a second-order critical transitions  -> transition

L. 399: represents a fundamental principles -> fundamental principle

L. 386: Zipf?s law -> Zipf's law

L. 554: as as the sum -> as the sum